# Autophagy-Dependent Reactivation of Epstein-Barr Virus Lytic Cycle and Combinatorial Effects of Autophagy-Dependent and Independent Lytic Inducers in Nasopharyngeal Carcinoma

**DOI:** 10.3390/cancers11121871

**Published:** 2019-11-26

**Authors:** Stephanie Pei Tung Yiu, Kwai Fung Hui, Christian Münz, Kwok-Wai Lo, Sai Wah Tsao, Richard Yi Tsun Kao, Dan Yang, Alan Kwok Shing Chiang

**Affiliations:** 1Department of Paediatrics and Adolescent Medicine, Li Ka Shing Faculty of Medicine, The University of Hong Kong, Queen Mary Hospital, Pokfulam, Hong Kong, China; stephanie.pty@gmail.com (S.P.T.Y.); Kfhui111@gmail.com (K.F.H.); 2Viral Immunobiology, Institute of Experimental Immunology, University of Zurich, CH-8006 Zurich, Switzerland; christian.muenz@uzh.ch; 3Department of Anatomical and Cellular Pathology, State Key Laboratory in Oncology in South China, Prince of Wales Hospital, The Chinese University of Hong Kong, Hong Kong, China; kwlo@cuhk.edu.hk; 4Li Ka Shing Institute of Health Science, The Chinese University of Hong Kong, Hong Kong, China; 5Department of Anatomy, Li Ka Shing Faculty of Medicine, The University of Hong Kong, Pokfulam, Hong Kong, China; gswtsao@hku.hk; 6Center for Nasopharyngeal Carcinoma Research, The University of Hong Kong, Hong Kong, China; 7Department of Microbiology, Li Ka Shing Faculty of Medicine, The University of Hong Kong, Hong Kong, China; rytkao@hkucc.hku.hk; 8Department of Chemistry, The University of Hong Kong, Hong Kong, China; yangdan@hku.hk

**Keywords:** autophagy, Epstein-Barr virus (EBV), lytic reactivation, nasopharyngeal carcinoma (NPC), lytic induction therapy, drug combination

## Abstract

Autophagy, a conserved cellular mechanism, is manipulated by a number of viruses for different purposes. We previously demonstrated that an iron-chelator-like small compound, C7, reactivates Epstein-Barr virus (EBV) lytic cycle by activating the ERK1/2-autophagy axis in epithelial cancers. Here, we aim to identify the specific stage of autophagy required for EBV lytic reactivation, determine the autophagy dependency of EBV lytic inducers including histone deacetylase inhibitor (HDACi) and C7/iron chelators, for EBV lytic reactivation and measure the combinatorial effects of these types of lytic inducers in nasopharyngeal carcinoma (NPC). Inhibition of autophagy initiation by 3-MA and autolysosome formation by chloroquine demonstrated that only autophagy initiation is required for EBV lytic reactivation. Gene knockdown of various autophagic proteins such as beclin-1, ATG5, ATG12, ATG7, LC3B, ATG10, ATG3 and Rab9, revealed the importance of ATG5 in EBV lytic reactivation. 3-MA could only abrogate lytic cycle induction by C7/iron chelators but not by HDACi, providing evidence for autophagy-dependent and independent mechanisms in EBV lytic reactivation. Finally, the combination of C7 and SAHA at their corresponding reactivation kinetics enhanced EBV lytic reactivation. These findings render new insights in the mechanisms of EBV lytic cycle reactivation and stimulate a rational design of combination drug therapy against EBV-associated cancers.

## 1. Introduction

Epstein-Barr virus (EBV), a γ-herpesvirus, infects over 90% of the adult human population worldwide and is associated with various cancers, including Burkitt’s lymphoma (BL), Hodgkin disease (HD), nasopharyngeal carcinoma (NPC) and a subset of gastric carcinoma (GC), of which NPC is highly prevalent in the regions of Southern China [1,2]. In these associated cancers, EBV remains in its latent form expressing only a few EBV proteins. In order to achieve EBV-targeted killing of these cancer cells, reactivation of the lytic cycle has been reported to efficiently induce destruction of these cells [3,4,5]. In a previous study, we have characterized the mode of action of a novel compound, C7, that reactivates EBV lytic cycle via intracellular iron chelation and activation of the ERK1/2-autophagy axis [6]. Given that autophagy is a conserved cellular mechanism for maintaining cellular homeostasis and plays an essential role in governing cell death and survival [7,8,9], we endeavor to further delineate the role of autophagy in EBV lytic reactivation.

Autophagy progression involves a number of sequential events that are tightly regulated. Autophagy induction in response to external stimuli, such as nutrient deprivation, activates the ATG1 kinase complex, i.e., ATG1, ATG13 (regulatory protein) and ATG17-ATG29-ATG31 (scaffold subcomplex), via the repression of TORC1 and/or activation of AMPK. This complex functions in recruiting other ATG proteins to the phagophore assembly sites (PAS) and activates downstream proteins by phosphorylation. Subsequently, a class III Ptdlns3K complex I (PI3KC3), which consists of Vps34 (lipid kinase), Vps15 (regulatory kinase), Vps30/Atg6, Atg14 and Atg38, is recruited to the PAS of the phagophore [10]. PI3 phosphorylation by this complex at the PAS allows for the recruitment of ATG18 and ATG2 which are required for the attachment of ATG8, ATG9 and ATG12 to the PAS. ATG12 and ATG8 represent two essential ubiquitin-like molecules required for autophagosome formation. ATG12 is conjugated to ATG5 via ATG7 (E1-like enzyme) and ATG10 (E2-like enzyme) [11,12]. The conjugate then binds with ATG16L to form a dimeric ATG12-ATG5-ATG16L E3-like enzyme for ATG8. This is then covalently attached to the lipid phosphatidylethanolamine (ATG8-PE) via the action of ATG4 (protease), ATG7 (E1-like enzyme), ATG3 (E2-like enzyme) and the dimeric ATG12-ATG5-ATG16L complex (E3-like enzyme). Lipidated ATG8 mediates membrane elongation, substrate recruitment and finally completion of the double membrane-surrounded autophagosome formation [13,14]. Afterwards, ATG8-PE is cleaved by ATG4 from the outer autophagosomal membrane to initiate the fusion between autophagosomes and lysosomes for substrate and inner autophagosome membrane degradation.

Viruses are known to manipulate the host’s autophagy machinery. In the context of EBV, interactions between EBV proteins including EBNA1 [15], EBNA3C [16], LMP1 [17,18], LMP2A [19] and Rta/Zta [20,21] with autophagy have been reported. EBNA1 was shown to be processed by autophagy and presented on MHC-II molecules in EBV-transformed B cells. Moreover, accumulation of EBNA1 could be observed in the autophagosomes of EBV-positive lymphoblastoid cells [15]. EBNA3C had been reported to be able to accelerate the transcription of autophagic genes i.e., *atg3*, *atg5* and *atg7*, as a survival mechanism in B cells [16], while LMP1 has been reported to initiate autophagy progression via the NF-κB pathway in B cells [17,18]. LMP2A could enhance the expression of autophagic proteins and facilitate autophagosome formation in epithelial cells [19].

In EBV lytic cycle induction, a study showed that the immediate early lytic protein Rta but not Zta, could initiate autophagy via the ERK1/2 signaling pathway. By inhibiting autophagy by 3-MA and atg5 knockdown abrogated the expression of EBV lytic proteins and the production of viral particles in B cells [20]. Another study showed that autophagic flux was blocked before the formation of the autolysosome during EBV lytic reactivation in B cells, and was hypothesized to be caused by the action of EBV late lytic proteins [21]. It was also suggested that the autophagic machinery was manipulated to limit the degradation of viral components and allow autophagic vesicles to facilitate virus packaging and release. Indeed, LC3B protein was found in the envelope of EBV viral particles, providing evidence to support the interplay between autophagy and the EBV life cycle [22]. In addition, reactivation of the EBV lytic cycle by TPA and sodium butyrate (T/B) was induced via the activation of PKC theta, p38 MAPK pathway and the initiation of autophagy in B cells [23].

Despite various reports on the interplay between autophagy and the EBV life cycle in B cells, little information on how autophagy reactivates the lytic cycle in EBV-associated epithelial cancers is available. In this study, we aim to identify the specific stage of autophagy required for EBV lytic reactivation, determine the autophagy dependency of EBV lytic inducers including histone deacetylase inhibitor (HDACi) and C7/iron chelators, for EBV lytic reactivation and measure the combinatorial effects of these types of lytic inducers in NPC. We hypothesized that certain stage or specific autophagic protein is involved in EBV lytic reactivation and that lytic activation by autophagy is only limited to certain class of lytic inducers. We also postulated that combining these drugs would increase the proportion of the cell population undergoing EBV lytic cycle which would, in turn, augment the specific killing of NPC cells.

## 2. Results

### 2.1. Endoplasmic Reticulum (ER) Stress Is Partially Required for Autophagy Initiation and EBV Lytic Reactivation

We have previously reported that C7 and iron chelators reactivate EBV lytic cycle by chelating intracellular iron and activating the ERK1/2-autophagy axis [6]. Given that autophagy can be induced by ER stress apart from iron chelation [24], the role of ER stress on EBV lytic reactivation upon C7 treatment is being studied. We first compared the expression of key ER stress markers, PERK and IRE-1α, in HA cells (a recombinant Akata EBV-infected NPC cell line) treated with either C7 or C7 pre-complexed with iron. We observed that the expression of both PERK and IRE-1α was not altered after 2 h of treatment in these two conditions, indicating that iron chelation did not rapidly induce ER stress in these cells (Figure 1a). In order to establish a chronological relationship between ER stress, autophagy and EBV lytic reactivation, their kinetics was assessed. We observed that in HA cells, the expression of ER stress markers, PERK, IRE-1α, BiP, XBP-1s and CHOP, followed a similar timeline with the EBV immediate early lytic protein, Zta, and lipidation of the autophagy marker LC3B (LC3B-II), one of the mammalian ATG8 orthologues (Figure 1b). This could also be observed in NPC43 and C666-1 cells (both NPC cell lines carrying native EBV genomes) (Figure 1c,d), suggesting that ER stress might be required for both EBV lytic reactivation and autophagy induction upon C7 treatment.

Next, siRNA-mediated silencing of PERK and IRE-1α was performed to verify the role of ER stress during EBV lytic reactivation and autophagy induction. Upon C7 treatment, we noticed there were discrepancies in the involvement of ER stress among different EBV+ NPC cells lines. In HA cells, the expression of Zta was reduced in either PERK or IRE-1α knockdown cells while that of LC3B-II was reduced only in PERK knockdown cells (Figure 1e). In C666-1 cells, the expression of Zta was reduced in either PERK or IRE-1α knockdown cells while that of LC3B-II was reduced only in IRE-1α knockdown cells (Figure 1f). In NPC43 cells, the expression of Zta remain unchanged despite the knockdowns while that of LC3B-II was reduced in either PERK or IRE-1α knockdown cells (Figure 1g). This suggested that ER stress is partially required for EBV lytic reactivation with preferences in ER stress pathway in HA via PERK-autophagy axis and in C666-1 via IRE-1α-autophagy axis. Since HA is not a native EBV-carrying cell, the discrepancies in the dependency on ER stress between HA and C666-1 cells may be affected by the biological differences in these two types of cells. For NPC43 cells, Rho kinase inhibitor (ROCKi, Y27632), which is required to maintain the EBV latency, may affect the involvement of ER stress in EBV lytic reactivation upon C7 treatment. Thus ER stress is only partially required for EBV lytic reactivation and autophagy induction in HA and C666-1 lines upon C7 treatment.

### 2.2. Autophagic Protein ATG5 Is Predominantly Required for EBV Lytic Reactivation upon C7 Treatment

Previously, we showed that autophagy is related to EBV lytic reactivation in epithelial cells [6]. In order to define the specific stage of autophagy responsible for this event, HA cells were treated with C7 or in combination with either 3-MA, a Vps34 inhibitor that blocks autophagy initiation, or with chloroquine, an autophagy inhibitor that inhibits autolysosome formation. We observed that Zta expression was abolished significantly (avgs. 53.8% vs 0.11%) in cells treated with the combination of C7 and 3-MA (Figure 2a) whereas in cells treated with the combination of C7 and chloroquine, the Zta expression remained unchanged (avgs 46.0% vs 49.8%) (Figure 2b). These suggested that only the initiation stage and stages before the maturation/degradation stage of autophagy are involved in EBV lytic reactivation.

The above result prompted us to hypothesize that only a certain stage or even a specific autophagic protein is required for EBV lytic reactivation. In order to test this hypothesis, autophagic proteins along the nucleation and elongation stages of the classical autophagy pathway were knocked down by siRNA. These cells were then treated with C7 and the expression level of Zta was measured.

We first performed siRNA knockdown of beclin-1, a key initiator of autophagy that is involved in membrane nucleation [10]. From our result, we could observe a significant reduction in the expression of Zta in beclin-1 knockdown HA cells (*p* = 0.0273) (Appendix A). This verified our hypothesis that autophagy initiation is required for EBV lytic reactivation. We then proceed to test autophagic proteins that are involved in the elongation stage of the autophagy pathway. siRNA knockdown was performed on ATG7, an E1-like activating enzyme [11]; ATG10, an E2-like conjugating enzyme [12]; LC3B, a key protein in autophagosome formation [13]; ATG3, an E2-like enzyme that conjugates LC3B to PE for driving autophagosome formation [25] and ATG5, an E3-like enzyme for LC3B lipidation [14]. From our results, knockdown of ATG7, ATG10, LC3B and ATG3 insignificantly reduced or did not affect the expression level of Zta, i.e., *p* = 0.4371; *p* = 0.8078; *p* = 0.617; *p* = 0.2061, respectively (Appendix A). Surprisingly, the expression of Zta was significantly compromised in ATG5 knockdown cells (*p* = 0.0086) (Figure 2c). We then knocked down ATG5 in NPC43 and C666-1 cells, and consistent with the results in HA cells, we could also observe a reduction in Zta expression (Figure 2d). Since most of the ATG5 proteins is consistently conjugated with ATG12 during the autophagy process [14], in order to determine if EBV lytic reactivation requires ATG5 protein alone or the ATG5-ATG12 conjugate, siRNA knockdown was also performed on ATG12. From our result, knockdown of ATG12 insignificantly reduced the expression level of Zta, i.e., *p* = 0.0775 (Appendix A), suggesting that ATG5 is solely required for EBV lytic reactivation upon C7 treatment.

Apart from the classical autophagy pathway, we also considered the alternative autophagy pathway in which autophagosomes originate from the Golgi apparatus [26]. siRNA knockdown was performed on a key protein, Rab9, of this pathway. The knockdown of Rab9 suppressed Zta expression in HA cells (*p* = 0.0448) at a barely significant level (Figure 2d). Moreover, we could not observe a strong reduction in Zta expression in Rab9-knockdown NPC43 and C666-1 cells (Figure 2f), suggesting that the involvement of Rab9 in EBV lytic reactivation is not as strong as that of ATG5. Taken together, the above knockdown experiments have identified the universally important role of ATG5 in EBV lytic reactivation in EBV-associated NPC cells upon C7 treatment.

### 2.3. Positive Feedback Loop between Zta Protein and Autophagy Induction

Since EBV and its viral proteins can manipulate various host cell mechanisms in B cells [15,16,17,18,19,20,21], we wonder about the mechanism of interaction between EBV proteins and autophagy in the epithelial cells. First, we compared autophagy progression by monitoring autolysosome formation in HA and HONE-1 cells (EBV-negative counterpart of HA). Given that autolysosomes are established by fusing autophagosomes with lysosomes [13], lysosomal marker, LAMP1, and autophagosome marker, LC3B, were stained with red and green fluorescent protein-tagged antibodies, respectively. Autolysosomes would appear as yellow puncta in fluorescent microscopy. From our results, within the first 30 h post-C7 treatment, autolysosomes can be observed in HA cells (Figure 3b, denoted with white triangles, *n* = 50 per time point). It spiked at 24 h post-C7 treatment in which autolysosomes can be observed in around 60% of the cells that underwent autophagy (Figure 3b and Appendix A). However, this could not be observed in HONE-1 cells. Autolysosome formation within the first 30 h post-C7 treatment is low i.e., around 1% on average (Figure 3a, *n* = 50 per time point), and it could only be first observed at 48 h post-C7 treatment (Appendix A, denoted with white triangles). The above result suggested that EBV could accelerate the progression of autophagy.

Moreover, when we compared autophagy initiation by measuring the appearance of LC3B puncta in the untreated HA and HONE-1 cells, spontaneous autophagy could only be observed in HA cells (Figure 3c, columns 3,4), indicating that EBV alone might initiate autophagy. To investigate the role of EBV in autophagy initiation, we transfected Zta-expressing plasmid into HONE-1 cells and the expression of Zta (in red) and formation of LC3B puncta (in green) were measured. We found that autophagy was initiated in Zta-transfected HONE-1 cells even without C7 treatment (Figure 3d, column 5), indicating that Zta alone could initiate autophagy. Together with the results from the previous section, initiation of autophagy precedes EBV lytic reactivation and EBV can accelerate autophagy progression through Zta, suggesting a positive feedback loop between autophagy initiation and EBV lytic reactivation, possibly for maintaining EBV lytic cycle.

### 2.4. Autophagy-Dependent and -Independent Reactivation of EBV Lytic Cycle by Lytic Cycle Inducers

In a previous study, we have demonstrated that C7/iron chelators reactivate EBV lytic cycle via the autophagy pathway [6] while the above sections further identified ATG5 as a key factor in EBV lytic reactivation and revealed a positive feedback loop between autophagy and EBV lytic cycle. In contrast, it has been reported that other pharmacologic lytic cycle inducers such as romidepsin and SAHA act through the PKC-δ pathway [27,28]. We thus hypothesize that autophagy-dependent EBV lytic reactivation is specific to C7/iron chelators and autophagy inhibition will only affect the lytic induction ability of C7/iron chelators but not on the other conventional lytic inducers. In order to test this hypothesis, we measured Zta expression via immunofluorescent staining in HA cells treated with C7, iron chelators such as deferoxamine, Dp44mT, deferiprone or conventional lytic inducers such as romidepsin, SAHA, in the absence and presence of 3-MA. First, we found that only C7 and the iron chelators could induce autophagy (as indicated by LC3B puncta formation) (Figure 4a). Second, inhibition of autophagy by 3-MA only affected Zta expression in cells treated with C7/iron chelators, i.e., column 3, 5, 7 and 9 while those by romidepsin or SAHA remained unaffected, i.e., columns 13 and 15 (Figure 4a).

This indicated that C7/iron chelators reactivate EBV lytic cycle via an autophagy-dependent pathway while romidepsin or SAHA do so via an autophagy-independent pathway. We can thus sub-categorize EBV lytic inducers into autophagy-dependent and independent subclasses. Apart from a distinct EBV reactivation mechanism, we also found that these two subclasses of compounds impose different effects on the host cells. Given that autophagy and ER stress could be initiated by reactive oxygen species (ROS) [29], we wonder whether ROS is induced. We found that C7 could significantly induce the release of cytosol superoxide (i.e., 57.1%), however, the same could not be detected in cells treated with romidepsin or SAHA (Figure 4b). We then tested if cytosol superoxide could reactivate EBV lytic cycle via ER stress-autophagy pathway. We first compared the level of cytosol superoxide in wildtype HA cells and PERK and IRE-1α knockdown HA cells. We found that there was no significant reduction in cytosol superoxide levels in these two conditions, indicating that ER stress is not upstream of cytosol superoxide (Figure 4c). We tested whether cytosol superoxide would lead to ER stress. We first showed that ROS scavenger Ebselen could effectively reduce cytosol superoxide induced by C7 in HA cells i.e., 35.6% to 9.29% (Figure 4d). We measured the expression of ER stress markers, PERK and IRE-1α, in HA cells treated with C7 in the absence or presence of Ebselen. We found that the expression levels of PERK, IRE-1α, Zta and LC3B remained unchanged in both conditions (Figure 4e). Moreover, C7 could not induce cytosol superoxide in NPC43 cells (Appendix A). These indicated that cytosol superoxide is unrelated to EBV lytic reactivation and its upregulation in HA cells was a side effect upon C7 treatment.

Apart from cytosol superoxide, the effects on cell cycle arrest upon treatment with different lytic inducers were also examined. HA cells were stained with propidium iodide after treatment with either C7, romidepsin or SAHA. Results showed that C7 mainly led to S phase arrest while both romidepsin and SAHA led to a non-S phase arrest as previously reported [30,31] (Figure 4f, upper panel). Similar findings were also observed in NPC43 cells (Figure 4f, lower panel). These results indicated that these lytic inducers have different mechanisms of EBV lytic cycle reactivation and induce different cellular responses.

### 2.5. Combinatorial Treatment of C7 and SAHA Results in Synergistic Killing of NPC Cells

We have reported previously [6,31] and further verified in this study that C7 or SAHA alone specifically killed EBV-positive HA cells relative to their EBV-negative counterpart, HONE-1 cells (Figure 5a,b). EBV has spontaneous lytic reactivation for progeny production while the extent this naturally-occurred lytic reactivation is kept at a low level in order to preserve cellular materials both for cell survival and production of viral proteins. Chemically reactivating EBV lytic cycle by C7 or SAHA may disrupt the homeostasis and result in excessive use of cellular molecules towards EBV lytic protein expression, causing a depletion of cellular materials which are important to sustain cell survival. Given that these lytic inducers reactivate EBV lytic cycle via different mechanisms and impose distinctive cellular effects, we hypothesized that combining inducers from each of the subclasses would increase the percentage of cells undergoing lytic cycle, thus augment the killing of EBV+ NPC cells. In order to test this hypothesis, we treated HA, C666-1 and NPC43 cells with C7 and SAHA simultaneously and measured their combinatorial effects via cell proliferation, degree of DNA fragmentation and the expression of Zta, cell death and apoptotic markers by flow cytometry. The combination of C7 and SAHA resulted in synergistic killing in HA (Figure 6c) and C666-1 cells (Figure 6d). Furthermore, the terminal deoxynucleotidyl transferase–mediated dUTP nick end labeling assay (TUNEL) showed that DNA fragmentation was more severe when HA (i.e., 53.5% in combination; 35.7% in C7; 13.7% in SAHA), NPC43 (18.7% in combination; 13.2% in C7; 12.1% in SAHA) and C666-1 (24.2% in combination; 14.6% in C7; 15.5% in SAHA) cells were treated with the combination of C7 and SAHA (Figure 5e). The above demonstrated that synergistic killing of EBV+ NPC cells could be achieved by the combination of autophagy-dependent and -independent lytic inducers.

We then measured the expression of Zta, LC3B puncta formation and cleaved caspase-3 (CC3, apoptosis marker) via immunofluorescent staining in HA cells upon treatment with C7, SAHA or the combination of the two lytic inducers. In C7-treated cells, we could observe a low expression of CC3 (15.9% in untreated; 22% in C7-treated cells) and a strong accumulation of LC3B puncta (0.01% in untreated; 10.9% in C7-treated cells). In SAHA-treated cells, only a strong signal of CC3 (i.e., 89.3%) could be observed. In cells treated with the combination of C7 and SAHA, a weaker expression of CC3 (i.e., 53.6%) and interestingly, a stronger accumulation of LC3B puncta could be observed (i.e., 38.9%) (Appendix A). Next, we quantified the percentage of aqua-blue (AB)-positive (indication of cell death) and the expression of CC3 and Zta by flow cytometry at different treatment times. In C7- treated cells at 24, 48 and 72 h, expression of Zta (avg. from AB/Zta and CC3/Zta staining) was in 8.52%, 9.68%, and 6.08% of cells while the expression of CC3 was observed in 8.6%, 16.8% and 20.6% of cells and the percentage of AB+ cells was 10.8%, 46.4% and 69.8%, respectively. In SAHA-treated cells, at 24, 48 and 72 h, 19.8%, 68.9%, and 68.4% of cells were expressing Zta (avg. from AB/Zta and CC3/Zta staining) while the expression of CC3 was found in 13.7%, 57.1% and 34.4% of cells and the percentage of AB+ cells was 19.6%, 90.01% and 86.6%, respectively. Lastly, in C7 and SAHA-double treated cells, at 24, 48 and 72 h, expression of Zta (avg. from AB/Zta and CC3/Zta staining) was in 11.4%, 18.2% and 19.6% of cells while the expression of CC3 was detected in 18.6%, 24.0% and 32.5% of cells and the percentage of AB+ cells was 40.6%, 81.5% and 89.9%, respectively. (Figure 5f and Appendix A).

We noticed that when we treated HA cells with C7 and SAHA simultaneously at their conventional treatment duration i.e., 48 h, the expression of CC3 was lower than the average percentage observed in cells treated with either C7 or SAHA alone, i.e., 24% vs 32.0% (Appendix A). For AB+ cells, the percentage was higher in cells treated with C7 and SAHA drug combination than the average percentage observed in cells treated with either C7 or SAHA alone for all the time points studied (Figure 5f), this suggested that enhanced cell death can be achieved by the combination of C7 and SAHA. Secondly, despite high percentage of cell death, AB-CC3-double-positive cells dropped from 50–70% in C7 or SAHA alone treated cells to 30–40% in cells treated with C7 and SAHA drug combination at 24 and 48 h while that maintained at similar percentage at 72 h. This suggested in addition to apoptosis, another cell death mechanism is involved in cells treated with the combined drug at 24 and 48 h. Given that enhanced accumulation of LC3B puncta was observed in cells treated with the combination of C7 and SAHA (Appendix A), we conjectured that cells were severely damaged due to the combined treatment and have converted autophagy from a protective mechanism to a suicide state, resulting in autophagic cell death. This potentially explained the non-apoptotic cell death enhancement observed in cells treated with the combined drug. Thirdly, Zta expression in cells treated with C7 and SAHA drug combination was lower than the average percentage observed in cells treated with either C7 or SAHA alone at all time points studied (Figure 5f).

We have also quantified the expression of Zta, CC3 and AB+ cells in NPC43 and C666-1 cells treated with C7 or SAHA alone or the combination of the two drugs for 72 h. For C7, SAHA and combination of C7 and SAHA treated NPC43 cells, expression of Zta (avg. from AB/Zta and CC3/Zta staining) was in 13.7%, 2.8%, and 1.6% of cells while the expression of CC3 was found in 41.5%, 21.3% and 21.5% of cells and the percentage of AB+ cells was 72.3%, 82.0% and 92.1%, respectively (Appendix A). For C7, SAHA and combination of C7 and SAHA treated C666-1 cells, expression of Zta (avg. from AB/Zta and CC3/Zta staining) was found in 5.9%, 1.7%, and 0.36% of cells while the expression of CC3 was observed in 4.6%, 5.1% and 6.4% of cells and the percentage of AB+ cells was 31.5%, 26.9% and 38.5%, respectively (Appendix A). Similar to that observed in HA cells, enhanced cell death could be achieved in NPC43 and C666-1 cells treated with combination of C7 and SAHA. The majority of cell death was caused by mechanisms other than apoptosis and the expression of Zta was lower in cells treated with combination of C7 and SAHA.

### 2.6. Combinatorial Treatment of C7 and SAHA at Their Corresponding Reactivation Kinetics Enhanced EBV Lytic Population

From the above section, despite the enhanced cell death found in cells treated with the combination of C7 and SAHA, unexpected reduction in Zta expression level was observed. In addition, only 15% of cells on average expressed both Zta and cell death marker, indicating that the majority of cell death was due to cellular mechanism unrelated to EBV lytic reactivation. This resulted in non-specific killing of both EBV+ and EBV- NPC cells. Indeed, synergism in cell death was detected in HONE-1 cells when treated with the combination of C7 and SAHA (Appendix A). In order to target the killing specifically to EBV+ NPC cells, we aimed to enhance the proportion of cells undergoing EBV lytic reactivation by matching their kinetics for lytic reactivation at the lowest dosage and shortening the treatment duration of C7 and SAHA. We noticed from our previous studies that the optimal treatment duration for SAHA was 48 h [31] while C7 could reactivate lytic cycle within 1 h of treatment.

We further monitored the reactivation kinetics of C7 and found that Zta expression spiked at 36 and 72 h post-C7-treatment (Figure 6b). Since prolonged C7 treatment has led to a high percentage of cell death (Figure 5f), we limited the treatment time of C7 to 24 h, which has only led to 11% of cell death from the previous result. This resulted in a sequential treatment scheme for SAHA and C7 (Figure 6c). In order to further limit non-specific cell death, a 2-fold reduction in doses of C7 and SAHA were used in this assay. Percentages of cell death, Zta and CC3 expression in HA cells treated with either C7 or SAHA alone, or sequentially with C7 and SAHA drug combination were measured by flow cytometry. For C7, SAHA and combination of C7 and SAHA treated HA cells, expression of Zta (avg. from AB/Zta and CC3/Zta staining) was found in 0.72%, 12.96%, and 17.85% of cells while the expression of CC3 was observed in 4.10%, 6.02% and 7.14% of cells and the percentage of AB+ cells was 7.73%, 48.5% and 43.3%, respectively (Figure 6d). Intriguingly, Zta expression has increased from 6.84% (the average percentage observed in cells treated with either C7 or SAHA alone) to 17.85% in cells sequentially treated with C7 and SAHA drug combination. Although the overall lytic population induced was not significantly high, this illustrated the importance of reactivation kinetics in lytic inducers combination therapy. Moreover, cells expressing both Zta and cell death marker had increased to 23%, suggesting that specific EBV+ cell death could also be achieved with this strategy.

Apart from SAHA, we have also tested the effects of combining C7 with romidepsin in HA, NPC43 and C666-1 cells. First, similar to SAHA, romidepsin alone conferred specific killing on EBV-positive HA cells compared to their EBV-negative counterpart, HONE-1 cells (Appendix A). Although the combination of C7 and romidepsin in HA (Appendix A) and C666-1 (Appendix A) resulted in synergistic killing, the dose at which the maximum synergism they could achieve required 52 and 37 μM, of C7, respectively, which was 8–9 fold higher than that found in the case of C7 and SAHA combination i.e., 5 μM, therefore, we doubted there was actual synergism in the C7 and romidepsin pair. Moreover, when we quantified the percentage of cell death and expression of CC3 and Zta by flow cytometry, no enhancement in cell death (or a slight increase at 24 h in HA cells), CC3 and Zta expression at all the time points studied in HA (Appendix A), NPC3 (Appendix A) and C666-1 (Appendix A) cells treated with C7 and romidepsin combination. This demonstrated that SAHA and romidepsin have different compatibility with C7, indicating that they are not interchangeable.

## 3. Discussion

The crosstalk between EBV and autophagy in B cells has been studied extensively in recent years, and EBV proteins, including EBNA1, EBNA3C, LMP1, LMP2A and Rta/Zta have been implicated in regulating autophagy initiation, progression and completion for EBV lytic reactivation, viral particle formation and release [6,7,8,9,10,11,12,13,14,15,16,17,18,19,20,21]. While a number of studies on the relationship between autophagy and EBV in B cells have been conducted, little information is available on the relationship between autophagy and EBV in epithelial cells. In a previous study, we have identified a novel compound, C7, from a high-throughput screening and demonstrated that it reactivates EBV lytic cycle via intracellular iron chelation and activation of the ERK1/2-autophagy axis in EBV-associated epithelial malignancies [6].

In this study, we aim to first define the specific stage of autophagy that is required for EBV lytic reactivation, determine the autophagy dependency of various lytic inducers of EBV including HDACi, C7/iron chelators and measure the combinatorial effects of these lytic inducers in NPC. By utilizing autophagy inhibitors, 3-MA and chloroquine, we showed that only autophagy initiation is required for EBV lytic reactivation (Figure 2). Through siRNA knockdown of various autophagic proteins, we found that only ATG5 knockdown significantly abrogated EBV lytic reactivation in HA, NPC43 and C666-1 cells (Figure 2 and Appendix A). However, autophagic proteins are interrelated to one another. Despite ATG5 is identified as a predominantly factor in EBV lytic reactivation, the involvement of other autophagic proteins cannot be completely excluded.

Next, we found that the presence of EBV could accelerate the kinetics for autolysosome formation (Figure 3). This finding was different from a previous report that when B cells were treated with bortezomib or TPA, autophagy progression was terminated at the stage by which autolysosome was formed [21]. We have shown in previous studies that bortezomib and SAHA induced EBV lytic cycle without the detection of autophagy initiation [6]. This is further supported in this study that SAHA adopted an autophagy-independent pathway in EBV lytic reactivation, hence, autophagy progression could not be determined in epithelial cells with these lytic inducers. Therefore, the discrepancies found between our study and the previous study [21] might be due to the biological differences between B cells and epithelial cells. In addition to the biological differences, another point to consider is the type of lytic inducers. In the B cell study, the author also hypothesized that the termination might be caused by EBV late lytic proteins. On the other hand, we have shown that C7 led to the formation of autolysosome but induced an abortive EBV lytic cycle without the expression of late lytic proteins in epithelial cells [6]. Since C7 is unable to reactivate EBV lytic cycle in B cells, new compounds capable of reactivating EBV lytic cycle in both B cells and epithelial cells will be required for a parallel study to further understand their biological differences in response to the lytic inducers.

In the second part of this study, we have categorized lytic inducers into autophagy-dependent (C7 and iron chelators) and autophagy-independent (HDACi) subclasses according to their mechanisms of action in EBV lytic reactivation. Given that pharmacologic lytic inducers have their own limitations in EBV lytic reactivation and killing [6], the discovery of autophagy-dependent and independent mechanisms of these lytic inducers prompted us to test their combinatorial effects in the NPC cells (Figure 4). Despite an enhanced killing achieved with C7 and SAHA drug combination, Zta expression was unexpectedly lower than the average percentage observed in HA, NPC43 and C666-1 cells treated with either C7 or SAHA alone at all time points studied (Figure 5). We have found that C7 and SAHA have different reactivation kinetics i.e., 48 h for SAHA and 36 and 72 h for C7, in which the discrepancies may potentially lead to antagonism. We therefore treated HA cells with C7 and SAHA in a sequential manner. Intriguingly, Zta expression level has increased (Figure 6). Despite a modest enhancement, it has pinpointed the importance of reactivation kinetics in drug combination study. On the other hand, cautions should be taken to avoid assumptions on compatibility of lytic inducers from the same category as shown in the antagonism observed with the C7 and romidepsin pair (Appendix A). A strategic schematic approach should therefore be implemented for rational design of drug combination in EBV lytic reactivation (Appendix A).

Manipulation of autophagy by different viruses is required in their replication process and autophagy is variably linked to different stages of cancer development [32]. As a result, pharmacologic modulation on autophagy opens up new concept of therapies particularly against virus-associated cancers [33]. For instance, adriamycin was shown to cause autophagic cell death in Hep3 cells by the continual activation of MAPK/ERK pathway [34,35]. Etoposide, a cytotoxic agent, might mediate cell death in HPV-associated cervical carcinoma by both apoptotic and autophagic cell death [36]. In this study, we showed that induction of autophagy reactivated EBV lytic cycle and led to cell death in EBV-positive NPC and combining compounds that modulate autophagy and those that cause apoptosis mediated synergistic killing of EBV-associated cancer cells. Further research on the role of autophagy in different viruses and cancer types may inform new strategies in manipulating this cellular mechanism in controlling viral infections as well as cancer progressions.

## 4. Materials and Methods

### 4.1. Cells and Culturing Conditions

HA cells (gifts from Prof. George S.W. Tsao of the University of Hong Kong, Hong Kong SAR, China) are a recombinant EBV infected nasopharyngeal carcinoma (NPC) cell line. The recombinant Akata EBV genomes of HA cells contain a neomycin resistant gene. HONE-1 cells (gifts from Prof. George S.W. Tsao) are the EBV-negative counterparts of HA cells. C666-1 is an EBV-positive NPC cell line that harbours native EBV genomes derived from an NPC xenograft of southern Chinese origin (gifts from Prof. George S.W. Tsao). NPC43 was established from patient NPC tissue (gifts from Prof. George S.W. Tsao). All the above cells were grown in RPMI 1640 media with 10% FBS at 37 °C, 5% CO_2_. HA and NPC43 cell lines are maintained in media with 500 μg/mL G418 and 4 μΜ Rho-associated coiled-coil containing kinases inhibitor (Y-27632), respectively. In order to prevent high background due to spontaneous lytic reactivation, Y-47632 was supplemented in media during lytic inducers treatment. All the cell lines used in this study have been authenticated by Genetica (Burlington, NC, USA) within three years and all experiments were performed with mycoplasma-free cells.

### 4.2. Cell Cycle Analysis

HA, C666-1 and NPC43 cells were seeded in 6-well cell culture plates. Cells at 70% confluence were either untreated or treated with either 20 or 40 μM C7, 5 or 10 nM romidepsin or 10 μM SAHA. After 12 h incubation, cells were first washed with PBS once and were subsequently fixed with 70% ethanol overnight at −20 °C. The next day, the cells were washed with PBS once and were resuspended and incubated in PBS supplemented with 500 μg/mL RNAase for 10 min at room temperature. The cells were then stained with 50 μg/mL of propidium iodide (Invitrogen, Carlsbad, CA, USA) in the dark at room temperature for 15 min. After staining, the cells were subjected to cellular DNA content analysis by a flow cytometer (LSRII, BD Biosciences (San Jose, CA, USA)). Data were analyzed by ModFit LT 3.0 software.

### 4.3. Drug Treatment

Unless otherwise specified, cells were treated at 70% confluence with the drugs at their specified concentrations for their designated treatment time depending on experimental needs in a 5% CO_2_, 37 °C incubator. Chemical compounds used: C7 (ID#5632947, ChemBridge, San Diego, CA, USA), romidepsin (S3020, Selleck, Houston, TX, USA), and suberoylanilide hydroxamic acid (SAHA, Cayman Chemical, Ann Arbor, MI, USA).

### 4.4. Immunofluorescent Staining

HONE-1 and HA cells grew on cover slips were treated with drugs for specific duration depending on experimental needs. Cells were first washed with PBS once and were fixed with ice-cold acetone for 10 min at room temperature. The fixed cells were then washed with PBS twice and were blocked with 5% normal goat serum (PCN5000, Gibco, Gaithersburg, MD, USA) in 1× TBST (50 mM Tris, 150 mM NaCl, 0.1% Triton-X, pH 7.4) for 30 min at room temperature. After that, the cells were stained with anti-Zta monoclonal antibody (1:50), LAMP-1, cleaved caspase-3, or LC3B rabbit polyclonal antibody (1:200; Cell Signaling Technology, Beverly, MA, USA) in 5% normal goat serum, 1× TBST overnight at 4 °C. Expression of the proteins was visualized with Alexa Fluor 594 F(ab′)2 fragment of goat anti-mouse IgG antibody or Alexa Fluor 488 F(ab′)2 fragment of goat anti-rabbit IgG antibody (1:500; Invitrogen) under fluorescence microscopy. Nuclei of cells were stained with 4′,6-diamidino-2-phenylindole (DAPI) (Roche, Mannheim, Germany).

### 4.5. Cytosol Superoxide Measurement

HA and NPC43 cells were seeded in 6-well plates. When the cells reached 70% confluence, culture media were removed and replaced with PBS. The cells were then incubated with 20 μM cytosol superoxide HKSOX-1r probe (gift from Prof. Dan Yang of the Department of Chemistry, The University of Hong Kong) for 30 min in a 37 °C, 5% CO_2_ incubator. After that, the cells were treated with 20 μM C7, 5nM romidepsin or 10 μM SAHA for 2 h. Cells were then collected and subjected to signal detection (FITC) by flow cytometry. Data were analyzed with FlowJo software.

### 4.6. MTT Assay

HONE-1, HA and C666-1 cells were seeded in 96-well plates. When the cells reached 70% confluence, they were either untreated or treated with or with 0, 0.625, 1.25, 2.5, 5, 10, 20, 40 μM C7; with 0, 0.312, 0.625, 1.25, 2.5, 5, 10, 20 μM SAHA for 48 h or with a gradient combination of C7 and SAHA for 72 h. A 3-(4,5-dimethylthiazol-2-yl)-2,5-diphenyltetrazolium bromide (MTT) solution was added into each well and wells were incubated in a 37 °C, 5% CO_2_ incubator for 5 h. OD430 nm and OD570 nm were measured and cell viability was plotted. To evaluate the synergistic action of C7 and SAHA, isobolograms were generated from the different combinations of concentrations of each drug which inhibit 50% of HA or NPC43 cells proliferation. Isoboles for IC50s that were located to the left of the additive isoboles indicated synergistic action. The combination index (CI) was calculated using the Chou and Talalay method. CI < 1, =1, and >1 represent synergy, additivity, and antagonism, respectively.

### 4.7. Transient Small Interference RNA and Short-Hairpin RNA Knockdown

siRNA targeting *atg5* (#M-004374-04-0005, Dharmacon, Lafayette, CO, USA), *atg7* (# M-020112-01-0005, Dharmacon), *beclin-1* (# M-010552-01-0005, Dharmacon,), *PERK* (# M-004883-03-0005, Dharmacon,), *IRE-1α* (#M-004951-02-0005, Dharmacon), *rab9* (# M-004177-01-0005, Dharmacon) and scramble siRNA (# D-001210-01-05, Dharmacon) were transfected with Lipofectamine 2000 (#11668027-, ThermoFisher, Waltham, MA, USA) according to the manufacturer’s protocol. Cells were incubated with transfection mix for 24 h, then the medium was removed. Cells were immediately treated with C7 for another 2 h. shRNA targeting LC3B (5’- AGATCGATCAGTTCATCTAAT-3’), ATG10 (5’-CAGCGTCCGAAGTGATTAAAT-3’), ATG3 (5’- CCTACCAACAGGCAAACAATT-3’) and ATG12 (5’-TGTGGGAGACACTCCTATTAT-3’), were cloned into pLOK.1 plasmid with EcoRI and AgeI restriction sites. Plasmids were verified with Sanger sequencing service (The University of Hong Kong). These plasmids were then transfected into HA cells by GeneJucie (MilliporeSigma, Burlington, MA, USA) with an incubation time of 24 h. The next day, medium were replaced with fresh medium and cells were treated with 20 μM C7 for 2 h. Experiments were repeated and the band intensities were quantified with ImageJ software. Differences in data were analyzed for statistical significance using unpaired Student’s t test. *P*-value less than 0.05 was considered as significant.

### 4.8. Terminal Deoxynucleotidyl Transferase–Mediated dUTP Nick End Labeling (TUNEL) Assay

HA, NPC43 and C666-1 cells were incubated with 20 μM C7, 10 μM SAHA or the combination of these drugs for 48 h (for HA cells); with 40 μM C7, 10 μM SAHA or the combination of these drugs for 72 h (for NPC43 and C666-1 cells). Following incubation, both floating and adherent cells were collected and washed twice with PBS. Terminal deoxynucleotidyl transferase-mediated dUTP nick end labeling (TUNEL) staining was then conducted with APO-BrdU TUNEL Assay Kit (Invitrogen, Waltham, MA, USA) following manufacturer’s instructions. The stained cells were detected by flow cytometry (LSRII; BD Biosciences) and data were analyzed by FlowJo software (Tree Star, Ashland, OR, USA).

### 4.9. Western Blot Analysis

Protein from the treated cell cultures was extracted and Western blot analysis was performed as described previously [37]. Expression of EBV lytic proteins was detected with anti-Zta (1:200; gift from Prof. P. Farrell, Imperial College, London, UK) and anti-BMRF1 (1:1000; gift from Dr.KH Chan, Department of Microbiology, HKU, Hong Kong SAR, China). Expression of phosphorylated ERK1/2 was detected with p-ERK1/2 rabbit polyclonal antibodies, (1:1000; #9101, Cell Signaling Technology Danvers, MA, USA). Expressions of ER stress markers PERK. IRE-1α, BiP, CHOP, XBP-1s was detected with ER stress sampler kit (1:1000; #9956, Cell Signaling Technology). Expression of autophagic proteins was detected with anti-ATG5 (1:1000; #12994, Cell Signaling Technology), anti-ATG7 (1:1000; #8558, Cell Signaling Technology,), anti-beclin-1 (1:1000; #3738, Cell Signaling Technology), anti-LC3B (1:1000; #2775, Cell Signaling Technology), anti-ATG10 (1:1000; #PA5-78593, Invitrogen), anti-ATG3 (1:1000; #3415, Cell Signaling Technology,), anti-ATG12 (1:1000; #4180, Cell Signaling Technology,) and anti-rab9 (1:1000; #5133. Cell Signaling Technology). Expression of human cellular α-tubulin and β-actin was detected with anti-α-tubulin and anti-β-actin antibody (1:5000; MilliporeSigma, Burlington, MA, USA), respectively, as loading controls.

### 4.10. Quantification of Zta and Cleaved Caspase-3 by Flow Cytometry

HA, C666-1 and NPC43 cells were allowed to grow to 70% confluence before treatment with the lytic cycle inducing compounds for different days depending on experimental needs. Cells were trypsinized and collected for each condition. They were then washed once with PBS, followed by staining with LIVE/DEAD Fixable Aqua Dead Cell Stain (1:320 in PBS) for 20 min on ice. Cells were washed once with PBS and were then fixed and permeabilized by the FACS fixation solution and FACS permeabilizing solution (BD Biosciences) for 30 min each. Cells were then stained with anti-cleaved caspase-3 rabbit monoclonal antibody (1:200 in 5% normal goat serum) and anti-Zta mouse monoclonal antibody (1:50 in 5% normal goat serum) overnight at 4 °C. The next day, cells were stained with Alexa Fluor 647 F(ab’)2 fragment of goat anti-mouse IgG (1:500; Life Technologies (Woburn, MA, USA)) and Alexa Fluor 488 F(ab’)2 fragment of goat anti-rabbit IgG at 37 °C for 1 h. The stained cells were then subject to analysis by flow cytometry (LSRII, BD Biosciences) and the data were analyzed with FlowJo software.

## 5. Conclusions

In conclusion, this study shows for the first time that autophagy initiation, in particular, the ATG5 protein is required for EBV lytic reactivation in NPC. C7/iron chelators and HDACi induce autophagy-dependent and independent mechanisms, respectively, to reactivate lytic cycle of EBV and impose differential cellular effects. Lastly, combination of C7 and SAHA at their corresponding reactivation kinetics enhances EBV lytic reactivation. This study provides the field with new insights on EBV pathogenesis by furthering our understanding of the mechanisms regulating EBV latent-lytic switch in epithelial cells and stimulates a rational design in drug combinations against EBV-associated cancers.

## Figures and Tables

**Figure 1 cancers-11-01871-f001:**
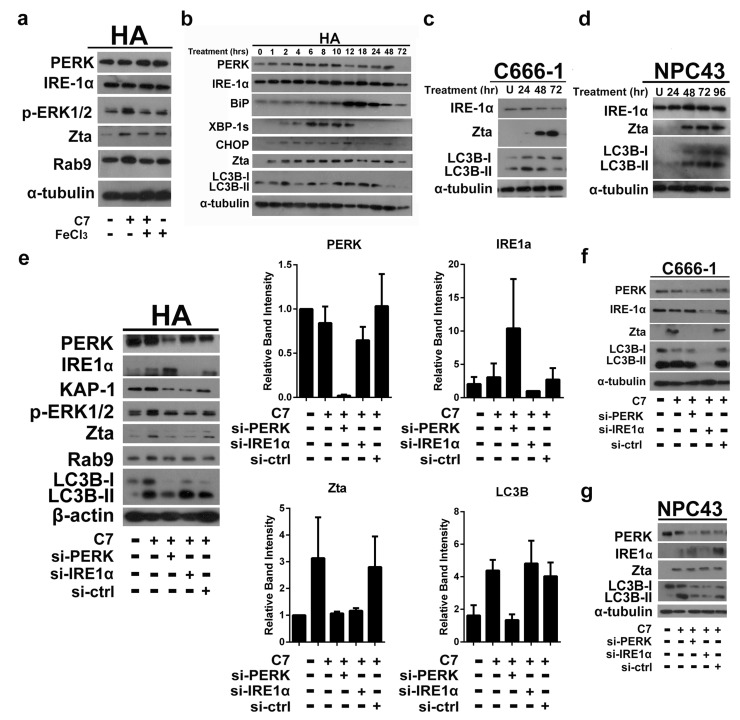
ER stress is required for autophagy initiation and EBV lytic reactivation. (**a**) HA cells were either untreated or treated with either 20 μM C7, 20 μM C7 precomplexed with iron or 20 μM FeCl_3_ alone for 2 h. The expression of PERK, IRE-1α, phosphorylated ERK1/2, Zta and Rab9 was detected by western blot analysis. Cellular α-tubulin was assessed as a loading control. (**b**) HA cells were treated with 20 μM C7 for 0, 1, 2, 4, 6, 8, 10, 12, 18, 24, 48 and 72 h. The expression of PERK, IRE-1α, BiP, XBP-1s, CHOP, Zta and LC3B was detected by western blot analysis. Cellular α-tubulin was used as a loading control. (**c**) C666-1 (**d**) NPC43 cells were either untreated or treated with 40 μM C7 for 24, 48 and 72 h. The expression of IRE-1α, Zta and lipidated or free LC3B (LC3B-II or LC3B-I, respectively) was detected by western blot analysis. Cellular α-tubulin was assessed as a loading control. (**e**) siRNA knockdown on PERK and IRE-1α was performed in HA cells. Scramble siRNA was performed as a control. The cells were treated with 20 C7 μM for 2 h. The expression of PERK, IRE-1α, KAP-1, phosphorylated ERK1/2, Zta, Rab9 and LC3B was detected by western blot analysis. Cellular β-actin was assessed as a loading control. Experiments were repeated and bands representing PERK, IRE-1α, Zta and LC3B were quantified and plotted as relative band intensity by ImageJ software and processed by GraphPad Prism 6 software. Statistics were calculated with Student’s T test. (**f**,**g**) siRNA knockdown on PERK and IRE-1α was performed in C666-1 and NPC 43 cells, respectively. Scramble siRNA was performed as a control. The cells were treated with 40 μM C7 for 72 h. The expression of PERK, IRE-1α, Zta, and LC3B was detected by western blot analysis. Cellular α-tubulin was assessed as a loading control.

**Figure 2 cancers-11-01871-f002:**
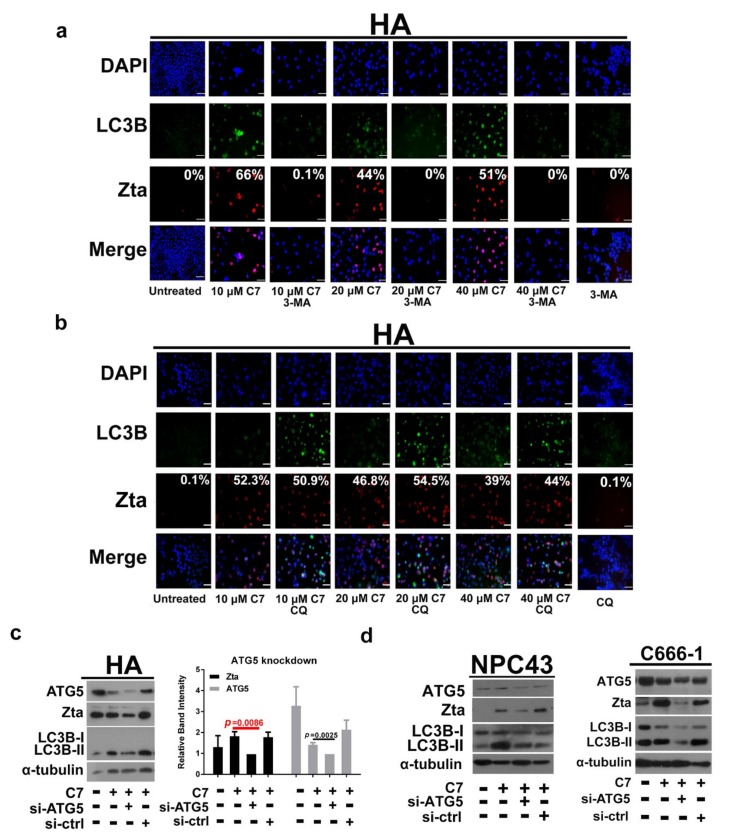
Autophagic protein ATG5 is predominantly required for EBV lytic reactivation upon C7 treatment. (**a**) HA cells were either untreated or treated with either 3-MA alone, 10, 20, 40 μM C7 or with these concentrations in combination with 3-MA for 48 h. Expression of Zta (red signals) and LC3B (green signals) was analyzed by immunofluorescence staining. DAPI (blue signals) stained cell nuclei. Scale Bar: 100 μm. (**b**) HA cells were either untreated or treated with either chloroquine alone, 10, 20, 40 μM C7 or with these concentrations in combination with chloroquine for 48 h. Expression of Zta (red signals) and LC3B (green signals) was analyzed by immunofluorescence staining. DAPI (blue signals) stained cell nuclei. Scale Bar: 100 μm. (**c**) siRNA knockdown on ATG5 was performed in HA cells. Scramble siRNA was performed as a control. The cells were treated with 20 C7 μM for 2 h. The expression of PERK, ATG5 and Zta was detected by Western Blot analysis. Cellular α -tubulin was assessed as a loading control. (**d**) siRNA knockdown of ATG5 was performed in NPC43 and C666-1 cells. Scramble siRNA was performed as a control. The cells were treated with 40 μM C7 for 72 h. The expression of ATG5, Zta and LC3B was detected by western blot analysis. Cellular α -tubulin was detected as a loading control. (**e**) siRNA knockdown of Rab9 was performed in HA cells. Scramble siRNA was performed as a control. The cells were treated with 20 C7 μM for 2 h. The expression of Rab9, Zta, Beclin-1 and LC3B (lipidated LC3B-II and free LC3B-I form) was detected by western blot analysis. Cellular β -actin was used as a loading control. (**f**) siRNA knockdown on Rab9 was performed in NPC43 and C666-1 cells. Scramble siRNA was performed as a control. The cells were treated with 40 μM C7 for 72 h. The expression of Rab9, Zta and LC3B was detected by western blot analysis. Cellular α -tubulin was assessed as a loading control. ATG5 and Rab9 siRNA knockdown experiments in HA cells were repeated in triplicates and bands representing Zta and the knockdown protein were quantified and plotted as relative band intensity. Statistics were calculated with Student’s T test.

**Figure 3 cancers-11-01871-f003:**
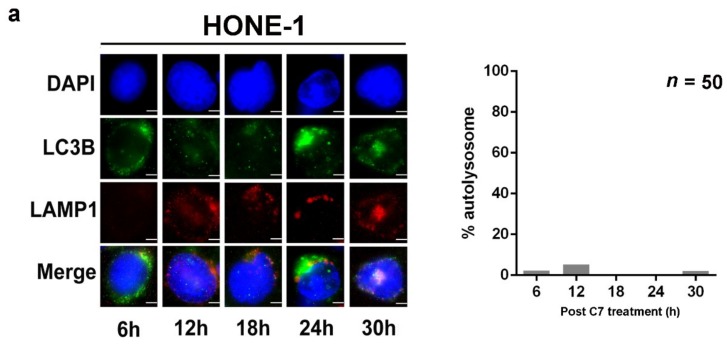
Positive feedback loop between Zta and autophagy initiation. (**a**) HONE-1 and (**b**) HA cells were treated with 20 μM C7 for 6, 12, 18, 24, or 30 h. Expression of lysosomal marker LAMP1 (red signals) and autophagosomes marker LC3B (green signals) was analyzed by immunofluorescence staining. DAPI (blue signals) stained cell nuclei. Scale Bar: 10 μm. Percentage of autolysosome formation were counted from 50 cells per time-point. (**c**) HONE-1 and HA cells were either untreated or treated with 20 μM C7 48 h. Expression of Zta (red signals) and LC3B (green signals) was analyzed by immunofluorescence staining. DAPI (blue signals) stained cell nuclei. Scale Bar: 100 μm. (**d**) HONE-1 cells transfected with pcDNA3 plasmid express Zta constitutively. Both Zta-transfected HONE-1 and wildtype HA cells were either untreated or treated with 20 μM C7 for 48 h. Expression of Zta (red signals) and LC3B (green signals) was analyzed by immunofluorescence staining. DAPI (blue signals) stained cell nuclei. Scale Bar: 100 μm.

**Figure 4 cancers-11-01871-f004:**
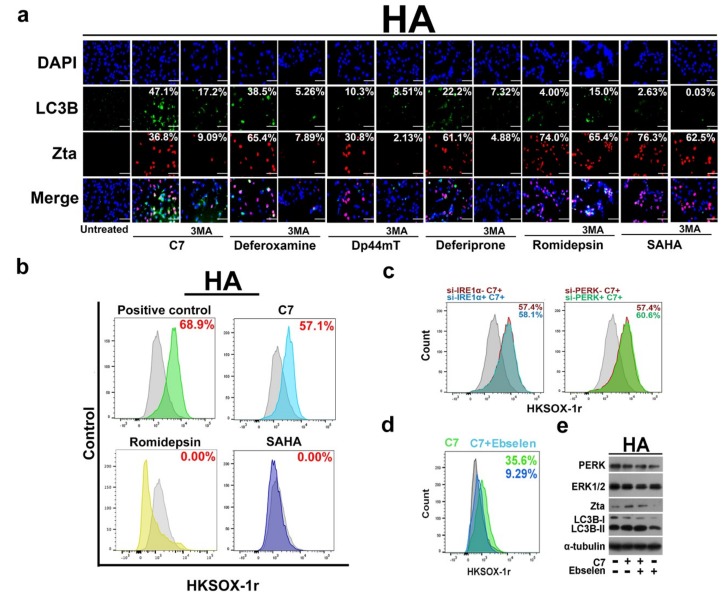
Autophagy-dependent and independent pathways are required for chemically induced EBV lytic reactivation. (**a**) HA cells were either untreated or treated with 20 μM C7, 1000 μM deferoxamine, 20 μM Dp44mT, 1000 μM deferiprone, 5 nM romidepsin, 10 μM SAHA or these compounds in combination with 3-MA for 48 h. Expression of Zta (red signals) and LC3B (green signals) was analyzed by immunofluorescence staining. DAPI (blue signals) stained cell nuclei. Scale Bar: 100 μm. (**b**) HA cells were pre-incubated with 2 μM HKSOX-1r cytosol superoxide probe for 30 min. The cells were then either untreated or treated with 20 μM C7, 5 nM romidepsin or 10 μM SAHA for 2 h. FITC signals were measured by flow cytometry and data were analyzed with FlowJo software. (**c**) siRNA knockdown of PERK and IRE-1α was performed in HA cells. Level of cytosol superoxide was measured as stated above upon 20 μM C7 treatment for 2 h. (**d**) HA cells were either treated with 20 μM C7 alone or in combination with 50 μM ROS scavenger, Ebselen, for 2 h. Level of cytosol superoxide was measured as stated above. (**e**) The HA cells in (**d**) was subjected to western blot analysis after flow cytometry measurement. The expression of PERK, IRE-1α, Zta and LC3B was detected by western blot analysis. Cellular α -tubulin was detected as a loading control. (**f**) HA and NPC43 cells were either untreated or treated with either 20 μM C7, 5 nM romidepsin or 10 μM SAHA for 12 h (HA cells); 48 h (NPC43 cells). Cell cycle arrest was analyzed, signals were measured by flow cytometry and data were analyzed with Modfit.3 software.

**Figure 5 cancers-11-01871-f005:**
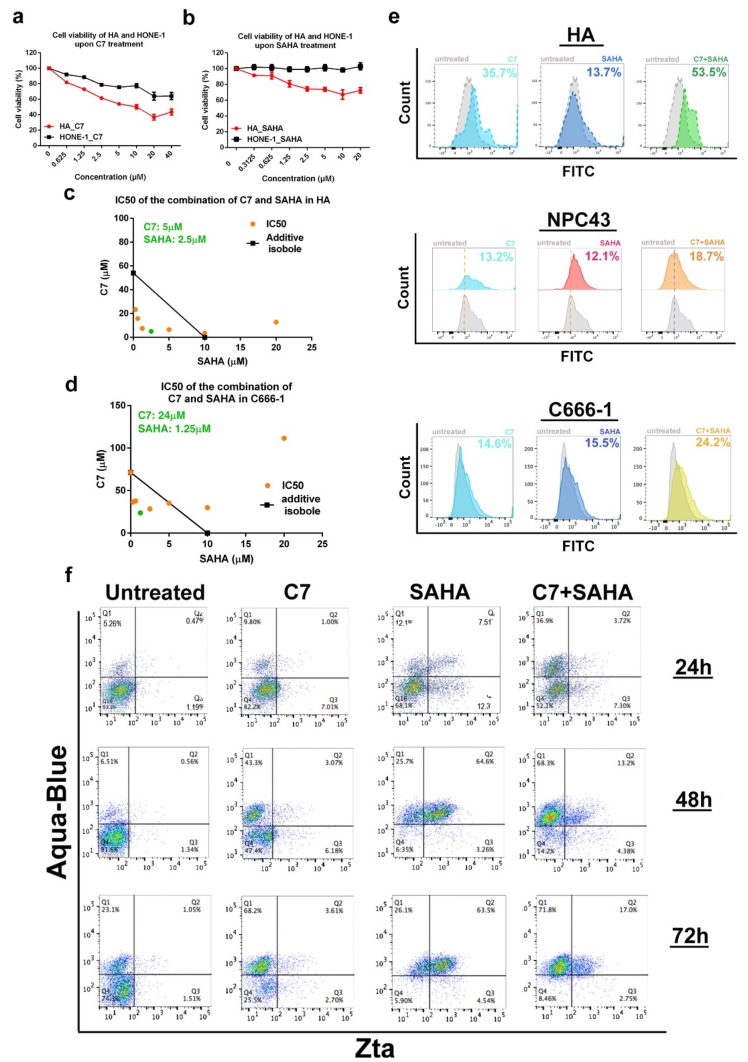
Combinatorial treatment with C7 and HDAC inhibitors result in synergistic killing of NPC cells. (**a**) HA and HONE-1 cells were treated with 0, 0.625, 1.25, 2.5, 5, 10, 20, or 40 μM C7 for 48 h followed by testing using MTT assay. Cell viability was plotted. (**b**) HA and HONE-1 cells were treated with 0, 0.312, 0.625, 1.25, 2.5, 5, 10, 20 μM SAHA for 48 h followed by testing with MTT assay. Cell viability was plotted. (**c**) HA and (**d**) C666-1 cells were treated with a gradient combination of C7 and SAHA for 48 h. Synergisms of proliferation inhibition of the two drugs in these cells were analyzed by isobologram analysis. (**e**) HA, NPC43 and C666-1 cells were either untreated or treated with 20 μM (40 μM for NPC43 and C666-1 cells) C7, 10 μM (20 μM for NPC43 and C666-1 cells) SAHA or the combination of the two compounds. DNA fragmentation was analyzed with TUNEL assay. Signals were detected by flow cytometry and data were processed with FlowJo software. (**f**) HA cells were either untreated or treated with 20 μM, 10 μM SAHA or the combination of the two compounds for 12, 24, 48, or 72 h. Percentage of cell death and the expression of CC3 and Zta was analyzed by flow cytometry as stated and data were analyzed with FlowJo software.

**Figure 6 cancers-11-01871-f006:**
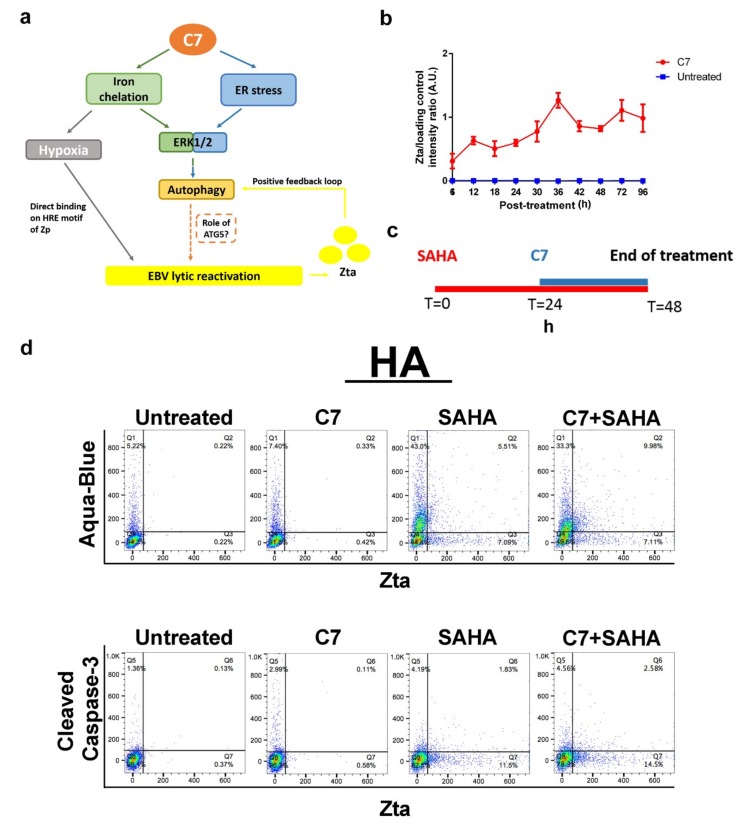
Combination of C7 and SAHA at their corresponding reactivation kinetics enhanced EBV lytic population. (**a**) Schematic diagram showing the mode of reactivation action of C7. (**b**) HA cells was treated with 20 μM C7 for 6, 12, 18, 24, 30, 36, 42, 48, 72 and 96 h. Expression of Zta was analyzed by western blot analysis. Band intensities for Zta and loading control were quantified with ImageJ and the intensity of Zta was then normalized with that of loading control. Error bar showed as SEM resulting from experimental triplicates. (**c**) Schematic diagram showing the treatment schedule for the sequential treatment of C7 and SAHA drug combination. (**d**) HA cells were either untreated or treated with 10 μM C7 for 24h, 5 μM SAHA for 48 h or the combination of the two compounds. Percentage of cell death and the expression of CC3 and Zta was analyzed by flow cytometry as stated and data were analyzed with FlowJo software.

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
