# Peer review of "Autophagy-Dependent Reactivation of Epstein-Barr Virus Lytic Cycle and Combinatorial Effects of Autophagy-Dependent and Independent Lytic Inducers in Nasopharyngeal Carcinoma"

_cancers, 2019, doi:10.3390/cancers11121871_

Round 1
Reviewer 1 Report
The manuscript submitted by Yiu e et al., described their observation on the complicated autophagy pathway and EBV reactivation. In the first part, the authors found the ER-stress markers are activated when the iron chelator C7 was used for virus reactivation. However, the results indicate the discrepancies of the knockdown effects on PERK and IRE-1 alpha in different cell lines. They also attempted to identified the key autophagy components that regulate EBV reactivation through the siRNA knockdown approach in different cell lines. ATG5 appeared to be important for EBV lytic reactivation in all different cell lines as indicated by the expression levels of Zta protein. In addition, Zta expression also induced autophagy activation and it was demonstrated by the co-expression of Zta and LC3B in various NPC derived cell lines. Different lytic cycle inducers were then put together for an attempt to see if any specific pathway involving autophagy could be identified, but without specific conclusion. In the last, the authors then tried to combine the effects of C7 with HDAC inhibitors to kill EBV positive cells, while cell death is not correlated with Zta expression.
Comments
Overall, this manuscript tried to address the correlation between autophagy pathway and Zta expression in different cell lines. However, it is very difficult to follow because of the lack of mechanism postulation and the significance of the final combination killing part is not clear.
Specific points
In this study, different EBV positive epithelial cell lines were used as model systems. The title “… in nasopharyngeal carcinoma” is not appropriate. A hypothesis-based study design provided in the beginning of the study is highly suggested. In the siRNA screening, ATG5 appeared to affect Zta expression in all different cell lines. However, the specificity of siATG5 was not confirmed by expression of siRNA resistant clone. Furthermore, it could be interesting if the mechanism could be identified. If it was reported that Zta expression along in Hone-1 cells initiate autophagy, it is difficult to dissect whether autophagy may be required for later stage of virus replication, rather than the initiation of Zta expression. The cell population showing co-staining signals in Fig. 3 should be indicated. The combination effects on cell killing is confusing. What is the rationale for the test? In line 360, “The majority of the cell death was caused by mechanisms other than apoptosis and expression of Zta was lower in cells treated with combination of C7 and SAHA.” So what is the mechanism of cell killing? It is not clear for the reason of kinetic studying of combined drug treatment. Do authors consider virus need to replicated? If the original thought was to kill EBV positive epithelial cells, it seems to be even better if no virus replication occurs. Especially if the HONE-1 parental cells do not die in the presence of combined drugs. What cause the EBV-specific killing? Overall, this reviewer suggests to put the manuscript into two different stories. The mechanism involved in ATG5 is interesting. The other part is to clarify the mechanism of EBV-mediated killing of combined drug treatment, which may lead to future clinical management of EBV positive tumors. An animal test would be required to draw a conclusion.

Author Response
Response to reviewer 1’s comments:
Comment:
“Different lytic cycle inducers were then put together for an attempt to see if any specific pathway involving autophagy could be identified, but without specific conclusion.”
Response:
Thank you for your comment, we apologize for not mentioning the hypothesis and conclusion clearly for this assay. We hypothesize that autophagy-dependent EBV lytic reactivation is specific to C7/iron chelators and autophagy inhibition will only affect the lytic induction ability of C7/iron chelators but not on the other conventional lytic inducers e.g. SAHA and Romidepsin. In order to test this hypothesis, we measured Zta expression via immunofluorescence staining in cells treated with C7, Deferoxamine, Dp44mT, Deferiprone, Romidepsin or SAHA alone or these drugs in combination with 3-MA. We found that inhibition of autophagy by 3-MA only affected Zta expression in cells treated with C7/iron chelators; while those by Romidepsin and SAHA remain unaffected. This indicated that C7/iron chelators reactivate EBV lytic cycle via an autophagy-dependent pathway while that for SAHA and Romidepsin, an autophagy-independent pathway is involved. We can thus subcategorize EBV lytic inducers into autophagy-dependent and independent subclass. In order to aid the delivery of this message, we have made amendments on page 10 line 268-286.
Amendments:
Page 10 line 268-286: “In a previous study, we have demonstrated that C7/iron chelators reactivate EBV lytic cycle via the autophagy pathway [6]; while the above sections further identified ATG5 as a key factor in EBV lytic reactivation and revealed a positive feedback loop between autophagy and EBV lytic cycle. In contrast, it has been reported that other pharmacologic lytic cycle inducers such as romidepsin and SAHA act through the PKC- pathway [27,28]. We thus hypothesize that autophagy-dependent EBV lytic reactivation is specific to C7/iron chelators and autophagy inhibition will only affect the lytic induction ability of C7/iron chelators but not on the other conventional lytic inducers. In order to test this hypothesis, we measured Zta expression via immunofluorescent staining in HA cells treated with C7, iron chelators i.e. deferoxamine, Dp44mT, deferiprone, or conventional lytic inducers i.e. romidepsin, SAHA, in the absence and presence of 3-MA. First, we found that only C7 and the iron chelators could induce autophagy (as indicated by LC3B puncta formation) (Figure 4a). Second, inhibition of autophagy by 3-MA only affected Zta expression in cells treated with C7/iron chelators, i.e. column 3, 5, 7 and 9; while those by romidepsin or SAHA remain unaffected, i.e. column 13 and 15 (Figure 4a). This indicated that C7/iron chelators reactivate EBV lytic cycle via an autophagy-dependent pathway while that for romidepsin or SAHA, an autophagy-independent pathway is involved. We can thus subcategorize EBV lytic inducers into autophagy-dependent and independent subclasses.”
Comment:
“However, it is very difficult to follow because of the lack of mechanism postulation and the significance of the final combination killing part is not clear.”
Response:
Thank you for your comment. As we have reported previously that C7 or SAHA alone specifically killed EBV-positive HA cells compared to their EBV-negative counterpart, HONE-1 cells. Since EBV naturally has spontaneous lytic reactivation for progeny production while the extent of this naturally-occurred lytic reactivation is kept low so as to balance cellular materials both for cell survival and the extra work on viral proteins production. By chemically reactivating EBV lytic cycle by C7 or SAHA, this may disrupt the homeostasis and resulted in the imbalance of cellular molecules towards extensive EBV lytic protein expression, consequently reducing cellular materials for sustaining cell survival and eventually resulted in cell death. Given that these lytic inducers reactivate EBV lytic cycle via different mechanism, we hypothesize that combining inducers from each of the subclass would enhance the percentage of cells undergoing lytic cycle, thus augmenting the killing of EBV+ NPC cells. In order to test this hypothesis, we combined C7 and SAHA to HA, C666-1 and NPC43 cells and measured their combinatorial effects via cell proliferation, degree of DNA fragmentation and the expression of cell death markers, apoptotic markers and lytic protein by flow cytometry.
From the above section, despite enhanced cell death in cells treated with the combination of C7 and SAHA, Zta expression was unexpectedly being reduced. Moreover, on average only 15% of cells expressing both Zta and cell death marker, indicating that majority of the cell death was due to cellular mechanism unrelated to EBV lytic reactivation. This may resulted in non-specific killing of both EBV+ and EBV- NPC cells. Indeed, synergism in cell death was resulted in HONE-1 cells when treated with the combination of C7 and SAHA (Supplementary Figure 7). In order to focus the killing to only EBV+ NPC cells, we aimed to enhance the proportion of cells undergoing EBV lytic reactivation by matching their kinetics for lytic reactivation at the same time limiting the drug toxicity on NPC cells by reducing the dosage and the treatment duration of C7 and SAHA.
The final combination with matching reactivation kinetics and the reduction in dosage and duration of C7 and SAHA has enhanced the proportion of cells undergoing lytic reactivation. Moreover, cells expressing both Zta and cell death marker has increased to 23%, suggesting that with this strategy, focused EBV+ cell death could be achieved.
We have made amendments regarding the first point on page 12 line 339-358; second point on page 15 line 437-445 and third point on page 15 line 466-468.
Amendments:
Page 12 line 339-358: “We have reported previously [6,31] and further verified in this study that C7 or SAHA alone specifically killed EBV-positive HA cells relative to their EBV-negative counterpart, HONE-1 cells (Figure 5a &b). EBV has spontaneous lytic reactivation for progeny production while the extent this naturally-occurred lytic reactivation is kept low so as to balance cellular materials both for cell survival and the extra work on viral proteins production. Chemically reactivating EBV lytic cycle by C7 or SAHA may disrupt the homeostasis and result in the imbalance of cellular molecules towards extensive EBV lytic protein expression, consequently reducing cellular materials for sustaining cell survival and eventually resulting in cell death. Given that these lytic inducers reactivate EBV lytic cycle via different mechanisms and imposed distinctive cellular effects, we hypothesized that combining inducers from each of the subclass would enhance the percentage of cells undergoing lytic cycle, thus augment the killing of EBV+ NPC cells. In order to test this hypothesis, we treated HA, C666-1 and NPC43 cells with C7 and SAHA simultaneously and measured their combinatorial effects via cell proliferation, degree of DNA fragmentation and the expression of Zta, cell death and apoptotic markers by flow cytometry.”
Page 15 line 437-445: “From the above section, despite the enhanced cell death found in cells treated with the combination of C7 and SAHA, unexpected reduction in Zta expression level was resulted. In addition, only 15% of cells on average expressed both Zta and cell death marker, indicating that the majority of cell death was due to cellular mechanism unrelated to EBV lytic reactivation. This resulted in non-specific killing of both EBV+ and EBV- NPC cells. Indeed, synergism in cell death was resulted in HONE-1 cells when treated with the combination of C7 and SAHA (Figure S7). In order to focus the killing specifically to EBV+ NPC cells, we aimed to enhance the proportion of cells undergoing EBV lytic reactivation by matching their kinetics for lytic reactivation at the lowest dosage and shortest available treatment duration of C7 and SAHA.”
Supplementary Figure 7.
Page 21 line 711: “Figure S7. Non-specific killing in both EBV+ and EBV- NPC cells. HONE-1 cells were treated with a gradient combination of C7 and romidepsin for 48 hrs. Synergisms of proliferation inhibition of the two drugs in these cells were analyzed by isobologram analysis.”
Page 15 line 466-468: “Moreover, cells expressing both Zta and cell death marker has increased to 23%, suggesting that with this strategy, focused EBV+ cell death could also be achieved.”
Comment:
“In this study, different EBV positive epithelial cell lines were used as model systems. The title “… in nasopharyngeal carcinoma” is not appropriate.”
Response:
HA cells are a recombinant EBV infected nasopharyngeal carcinoma (NPC) cell line. The recombinant Akata EBV genomes of HA cells contain a neomycin resistant gene. HONE-1 cells are the EBV-negative counterparts of HA cells. C666‐1 is an EBV‐positive NPC cell line that harbors native EBV genomes derived from an NPC xenograft of southern Chinese origin. NPC43 was established from patient NPC tissue. They are all NPC cell lines. Moreover, we have unpublished data that in EBV-associated gastric carcinoma cells, the relationship between autophagy and EBV lytic reactivation is not the same, therefore, to be precise, we stated “nasopharyngeal carcinoma” in the title.
Comment:
“A hypothesis-based study design provided in the beginning of the study is highly suggested.”
Response:
Thank you for your suggestion, we have made amendments on page 3 line 101-105 regarding this point.
Amendments:
Page 3 line 101-105: “We hypothesized that certain stage or specific autophagic protein is involved in EBV lytic reactivation, and that by autophagy is only limited to certain class of lytic inducers. We also postulated that combining these drugs would result in the enhancement of cell population undergoing EBV lytic cycle which consequently augment specific killing to EBV+ NPC cells.”
Comment:
“The specificity of siATG5 was not confirmed by expression of siRNA resistant clone.”
Response:
Thank you for your suggestion, we agree that using a RISC-free control would be nice for examining siRNA specificity. However, as shown in the western blot result in Figure 2, the intensities of the band representing ATG5 were significantly reduced in cells treated with si-ATG5; while that for cells treated with the scramble siRNA control, the intensity was comparable to that of untreated cells. This can be observed in HA, C666-1 and NPC43 cells. Moreover, we have repeated the above experiment at least three times in HA cells and from the relative intensity graph shown in Figure 2, the bar representing ATG5 signals from cells treated with si-ATG5 was significantly reduced while that from cells treated with the scramble siRNA control did not. We have also repeated the experiments twice in C666-1 and NPC43 cells, similar results to the one shown in Figure 2 could be obtained.
Comment:
“it could be interesting if the mechanism could be identified”
Response:
We agree that it would be interesting to completely dissect the mechanism of EBV lytic reactivation via the autophagy machinery. We have preliminary data showing that the region between -99 to -134 of the Zp is being upregulated upon C7 treatment in a luciferase study and we hypothesized that it is related to MEF2D as this transcription factor has been shown to be degraded by the autophagic machinery (doi: 10.1126/science.11660880). We are hoping to publish those data in a future manuscript as improvements need to be made on the system and more sophisticated experimental setups are needed to further confirm the findings.
Comment:
“If it was reported that Zta expression along in Hone-1 cells initiate autophagy, it is difficult to dissect whether autophagy may be required for later stage of virus replication, rather than the initiation of Zta expression”
Response:
Thank you for your question. We have shown previously that C7 induced an abortive EBV lytic cycle in which the expression of late lytic proteins e.g. gp350 and VCA-p18 could not be observed (https://doi.org/10.1371/journal.pone.0145994). Moreover, Raji cells could not be infected with supernatant collected from EBV+ HA cells induced by C7 in a transfection assay (doi: 10.3390/cancers10120505.). Since the virus replication is abrogated, we therefore considered that the upregulation of autophagy resulted from Zta expression was required for the positive feedback loop in maintaining the lytic cycle by continuously initiating Zta expression.
Comment:
“The cell population showing co-staining signals in Fig. 3 should be indicated.”
Response:
Thank you for your suggestion, the cell population of both HA and HONE-1 in Figure 3 has been added to the main figure and amendments have been made on page 9 line 233-242 and page 21 line 698. Additional figure was added to supplementary figure 2.
Amendments:
Page 8: Figure 3
Page 9 line 233-242: “From our results, within the first 30 hrs post C7 treatment, autolysosomes can observed in HA cells (Figure 3b, denoted with white triangles, n=50 per time point). It spiked at 24 hrs post-C7 treatment in which autolysosomes can be observed in around 60% of the cells that underwent autophagy (Figure 3b & Figure S2b). However, this could not be observed in HONE-1 cells. Autolysosome formation within the first 30 hrs post C7 treatment is low i.e. around 1% on average (Figure 3a, n=50 per time point), and it could only be first observe at 48 hrs post C7 treatment (Figure S2a, denoted with white triangles). The above result suggested that EBV could accelerate the progression of autophagy.”
Page 9 line 257-260: “Figure 3. Positive feedback loop between Zta and autophagy initiation. a) HONE-1 and b) HA cells were treated with 20 μM C7 for 6, 12, 18, 24, or 30 hrs. Expression of lysosomal marker LAMP1 (red signals) and autophagosomes marker LC3B (green signals) was analyzed by immunofluorescence staining. DAPI (blue signals) stained cell nuclei. Scale Bar: 250 μm. Percentage of autolysosome formation were counted from 50 cells per time-point.”
Supplementary Figure 2:
Page 21 line 722-725: “Figure S2. Positive feedback loop between Zta and autophagy initiation. a) HONE-1 and b) HA cells were treated with 20 μM C7 for 36, 42, 48, 72 or 96 hrs. Expression of lysosomal marker LAMP1 (red signals) and autophagosomes marker LC3B (green signals) was analyzed by immunofluorescence staining. DAPI (blue signals) stained cell nuclei. Scale Bar: 250 μm.”
Comment:
“The combination effects on cell killing is confusing. What is the rationale for the test? In line 360, “The majority of the cell death was caused by mechanisms other than apoptosis and expression of Zta was lower in cells treated with combination of C7 and SAHA.” So what is the mechanism of cell killing?”
Response:
Thank you for your comment. As mentioned in response 2, we have reported previously that C7 or SAHA alone specifically killed EBV-positive HA cells compared to their EBV-negative counterpart, HONE-1 cells. Since EBV naturally has spontaneous lytic reactivation for progeny production while the extent this naturally-occurred lytic reactivation is kept low so as to balance cellular materials both for cell survival and the extra work on viral proteins production. By chemically reactivating EBV lytic cycle by C7 or SAHA, this may disrupt the homeostasis and resulted in the imbalance of cellular molecules towards extensive EBV lytic protein expression, consequently reducing cellular materials for sustaining cell survival and eventually resulted in cell death. Given that these lytic inducers reactivate EBV lytic cycle via different mechanism, we hypothesize that combining inducers from each of the subclass would enhance the percentage of cells undergoing lytic cycle, thus augment the killing of NPC cells. In order to test this hypothesis, we combined C7 and SAHA to HA, C666-1 and NPC43 cells and measured their combinatorial effects via cell proliferation, degree of DNA fragmentation and the expression of cell death markers, apoptotic markers and lytic protein by flow cytometry. We have made amendments regarding this point on page 12 line 339-358.
For the mechanism of cell killing, we believed that autophagic cell death was involved in addition to apoptosis. In cells treated with the combination of C7 and SAHA, the apoptotic marker cleaved caspase-3 did not increased with the enhanced cell death, we therefore conjectured that in addition to apoptosis, other cell death mechanism was involved and that could be autophagic cell death as higher autophagy expression could be observed in cells treated with the combined drugs. We believed that when the cell is highly stressed by the introduction of both of the lytic inducers, autophagy will switched from its protective mode to its killing mode, consequently leading to autophagic cell death. To aid delivery of this message, we have made amendments on page 13, line 395-401.
Amendments:
Page 12 line 339-358: “We have reported previously [6,31] and further verified in this study that C7 or SAHA alone specifically killed EBV-positive HA cells relative to their EBV-negative counterpart, HONE-1 cells (Figure 5a &b). EBV has spontaneous lytic reactivation for progeny production while the extent this naturally-occurred lytic reactivation is kept low so as to balance cellular materials both for cell survival and the extra work on viral proteins production. Chemically reactivating EBV lytic cycle by C7 or SAHA may disrupt the homeostasis and result in the imbalance of cellular molecules towards extensive EBV lytic protein expression, consequently reducing cellular materials for sustaining cell survival and eventually resulting in cell death. Given that these lytic inducers reactivate EBV lytic cycle via different mechanisms and imposed distinctive cellular effects, we hypothesized that combining inducers from each of the subclass would enhance the percentage of cells undergoing lytic cycle, thus augment the killing of EBV+ NPC cells. In order to test this hypothesis, we treated HA, C666-1 and NPC43 cells with C7 and SAHA simultaneously and measured their combinatorial effects via cell proliferation, degree of DNA fragmentation and the expression of Zta, cell death and apoptotic markers by flow cytometry.”
Page 13 Line 395-401: “This suggested in addition to apoptosis, another cell death mechanism is involved in cells treated with the combined drug at 24 and 48 hrs. Given that enhanced accumulation of LC3B puncta was observed in cells treated with the combination of C7 and SAHA (Figure S4), we conjectured that cells were severely damaged due to the combined treatment and have converted autophagy from a protective mechanism to a suicide state, resulting in autophagic cell death. This potentially explained the non-apoptotic cell death enhancement observed in cells treated with the combined drug.”
Comment:
“It is not clear for the reason of kinetic studying of combined drug treatment. Do authors consider virus need to replicated? If the original thought was to kill EBV positive epithelial cells, it seems to be even better if no virus replication occurs.”
Response:
Thank you for your comment. Similar to response 2, although the result from Figure 5 showed enhanced cell death in cells treated with the combination of C7 and SAHA, Zta expression was unexpectedly being reduced. Moreover, on average only 15% of cells expressing both Zta and cell death marker, indicating that majority of the cell death was due to cellular mechanism unrelated to EBV lytic reactivation. This may resulted in non-specific killing of both EBV+ and EBV- NPC cells. Indeed, synergism in cell death was resulted in HONE-1 cells when treated with the combination of C7 and SAHA (Supplementary Figure 7). In order to focus the killing to only EBV+ NPC cells, we aimed to enhance the proportion of cells undergoing EBV lytic reactivation by matching their kinetics for lytic reactivation at the same time limiting the drug toxicity on NPC cells by reducing the dosage and the treatment duration of C7 and SAHA.
The final combination with matching reactivation kinetics, reduced dosage and duration of C7 and SAHA has enhanced the proportion of cells undergoing lytic reactivation. Moreover, cells expressing both Zta and cell death marker has increased to 23%, suggesting that with this strategy, focused EBV+ cell death could be achieved. We have made amendments on page Page 15 line 437-445; Page 15 line 466-468 and Page 21 line 711.
Amendments:
Page 15 line 437-445: “From the above section, despite the enhanced cell death found in cells treated with the combination of C7 and SAHA, unexpected reduction in Zta expression level was resulted. In addition, only 15% of cells on average expressed both Zta and cell death marker, indicating that the majority of cell death was due to cellular mechanism unrelated to EBV lytic reactivation. This resulted in non-specific killing of both EBV+ and EBV- NPC cells. Indeed, synergism in cell death was resulted in HONE-1 cells when treated with the combination of C7 and SAHA (Figure S7). In order to focus the killing specifically to EBV+ NPC cells, we aimed to enhance the proportion of cells undergoing EBV lytic reactivation by matching their kinetics for lytic reactivation at the lowest dosage and shortest available treatment duration of C7 and SAHA.”
Supplementary Figure 7.
Page 21 line 711: “Figure S7. Non-specific killing in both EBV+ and EBV- NPC cells. HONE-1 cells were treated with a gradient combination of C7 and romidepsin for 48 hrs. Synergisms of proliferation inhibition of the two drugs in these cells were analyzed by isobologram analysis.”
Page 15 line 466-468: “Moreover, cells expressing both Zta and cell death marker has increased to 23%, suggesting that with this strategy, focused EBV+ cell death could also be achieved.”
Comment:
“Especially if the HONE-1 parental cells do not die in the presence of combined drugs. What cause the EBV-specific killing?”
Response:
Thank you for your question. Given that EBV naturally has spontaneous lytic reactivation for progeny production while the extent this naturally-occurred lytic reactivation is kept low so as to balance cellular materials both for cell survival and the extra work on viral proteins production. By chemically reactivating EBV lytic cycle by lytic inducers e.g. C7 and SAHA, we conjectured that this may disrupt the homeostasis and resulted in the imbalance of cellular molecules towards extensive EBV lytic protein expression, consequently reducing cellular materials for sustaining cell survival and eventually resulted in cell death.
Comment:
“Overall, this reviewer suggests to put the manuscript into two different stories. The mechanism involved in ATG5 is interesting. The other part is to clarify the mechanism of EBV-mediated killing of combined drug treatment, which may lead to future clinical management of EBV positive tumors. An animal test would be required to draw a conclusion.”
Response:
Thank you for your suggestion. In the first part of the manuscript, we have identified a novel mechanism involving autophagy, specifically ATG5, in EBV lytic reactivation. This finding provide new insights to the field on the basic biology of EBV in manipulating a well-conserved cellular pathway. We therefore would like to translate this knowledge to EBV-targeted therapy. With the discovery of autophagy-dependent and independent lytic inducers, we believed that combining drugs from each of the subclasses could enhance the proportion of cells undergoing EBV lytic cycle, tilting the cellular materials towards extensive lytic protein production, eventually augmenting specific killing to EBV+ NPC cells. We think it would be a comprehensive story only if we combine both parts together, from the identification of a novel lytic reactivation mechanism to the translational potential of this finding. Therefore, we would like to present the story in this way.
We also agree that an animal test would be required for drawing a conclusion on the tumour reduction effects of the combination therapy in the future. At this stage, we aimed to introduce this idea of rational drug combination i.e. matching lytic kinetics to the field. In the future, more sophisticated and in-depth study regarding the combination therapy would be performed and an animal study would definitely be essential.

Reviewer 2 Report
The authors describe EBV reactivation that is mediated by dependent and independent pathway. The data presented in each experiment is interesting. However, the story they present here is too complicated to confirm the value of this paper.
How do they confirm abortive EBV lytic infection in various autophagy modulator treated experiments? The early and/or late EBV gene expression was not determined in the paper.
What the cell death mechanism that is not mediated via apoptosis? How the autophagy modulation the authors used cause these NPC cell killing.
How do we apply the result in the understanding of NPC carcinogenesis or in the NPC treatment?
Author Response
Response to reviewer 2’s comments:
Comment:
“The authors describe EBV reactivation that is mediated by dependent and independent pathway. The data presented in each experiment is interesting. However, the story they present here is too complicated to confirm the value of this paper.
Response:
Thank you very much for your comment, we apologize for confusing readers on our objectives for this manuscript. The main objective of this manuscript is to expand the reservoir of drug available for EBV-targeted therapy. We are particularly interested in the EBV lytic induction therapy which involved two components i.e. EBV lytic inducer and anti-viral drug e.g. ganciclovir. EBV is induced into lytic cycle by chemical inducers which activate GCV that consequently result in killing to EBV+ cells. However, EBV reactivation by conventional lytic inducers were highly depending on cellular background and not all EBV+ cell types could be induced, which limits the efficacy of the lytic induction therapy. Therefore, we previously performed a high throughput screening for novel compounds that induce EBV lytic cycle in hope to find compounds that have higher lytic induction ability or compounds use different lytic induction mechanism (doi: 10.1371/journal.pone.0145994). We have selected C7 for further study due to its ability in inducing multiple EBV+ epithelial cancers and its fast induction kinetic. We then found that C7 reactivates EBV lytic cycle via iron chelation and the ERK1/2-autophagy axis (doi: 10.3390/cancers10120505).
Given that autophagy is a conserved cellular mechanism for maintaining cellular homeostasis and plays an essential role in governing cell death and survival, while different viruses have been shown to manipulate this machinery for viral replication and particles production. In EBV, several viral proteins e.g. EBNA1, LMP1 and Rta, have been shown to interact with the autophagy machinery. However, most of these studies were performed in EBV+ B cells while little is known on the relationship between autophagy and EBV+ epithelial cells. Therefore, this manuscript aims to further delineate the mechanism of which autophagy reactivates EBV lytic cycle by C7 in EBV+ NPC cells. Through the knockdown assays, we identified ATG5 as a predominant factor required for EBV lytic induction in NPC cells. This is a novel finding to link autophagy with EBV lytic reactivation and to specifically identify an autophagic protein that is responsible for EBV lytic reactivation.
We hypothesized that EBV lytic induction by autophagy is specific to C7 and iron chelators, therefore we examined the autophagy-dependency on conventional lytic inducers, which then allowed us to sub-categorize these lytic inducers into autophagy-dependent and independent classes. This is the first study to report the autophagy dependency of the conventional lytic inducers and with this information, it allows follow-up studies on testing their combinatorial effect on lytic induction and EBV-targeted killing.
Since EBV naturally has spontaneous lytic reactivation for progeny production while the extent of this naturally-occurred lytic reactivation is kept low so as to balance cellular materials both for cell survival and the extra work on viral proteins production. By chemically reactivating EBV lytic cycle by C7 or SAHA, this may disrupt the homeostasis and resulted in the imbalance of cellular molecules towards extensive EBV lytic protein expression, consequently reducing cellular materials for sustaining cell survival and eventually resulted in cell death. Since these inducers reactivate EBV lytic cycle via different mechanism, we hypothesized that combining inducers from each of the subclass would enhance the percentage of cells undergo lytic cycle, thus augmenting the killing of EBV+ NPC cells. In order to test this hypothesis, we combined C7 and SAHA to HA, C666-1 and NPC43 cells and measured their combinatorial effects via cell proliferation, degree of DNA fragmentation and the expression of cell death markers, apoptotic markers and lytic protein by flow cytometry.
From the above section, despite enhanced cell death in cells treated with the combination of C7 and SAHA, Zta expression was unexpectedly being reduced. Moreover, on average only 15% of cells expressing both Zta and cell death marker, indicating that majority of the cell death was due to cellular mechanism unrelated to EBV lytic reactivation. This may resulted in non-specific killing of both EBV+ and EBV- NPC cells. Indeed, synergism in cell death was resulted in HONE-1 cells when treated with the combination of C7 and SAHA (Supplementary Figure 7). In order to focus the killing to only EBV+ NPC cells, we aimed to enhance the proportion of cells undergoing EBV lytic reactivation by matching their kinetics for lytic reactivation at the same time limiting the drug toxicity on NPC cells by reducing the dosage and the treatment duration of C7 and SAHA.
The final combination with matching reactivation kinetics and the reduction in dosage and duration of C7 and SAHA has enhanced the proportion of cells undergoing lytic reactivation. Moreover, cells expressing both Zta and cell death marker has increased to 23%, suggesting that with this strategy, focused EBV+ cell death could be achieved.
This is a study combining the identification of a novel mechanism for EBV lytic reactivation to translating this class of drugs into EBV-targeted therapy. This study could provide the field with new insights on EBV pathogenesis by understanding the mechanism regulating EBV latent-lytic switch and stimulate a rational design in drug combinations against EBV-associated cancers.
To aid the delivery of the story and the value of this manuscript, we have added some points throughout the manuscript, please see Page 12 line 339-358; Page 15 line 437-445; Page 21 line 711; Page 15 line 466-468; Page 21 line 695-702.
Amendments:
Page 12 line 339-358: “We have reported previously [6,31] and further verified in this study that C7 or SAHA alone specifically killed EBV-positive HA cells relative to their EBV-negative counterpart, HONE-1 cells (Figure 5a &b). EBV has spontaneous lytic reactivation for progeny production while the extent this naturally-occurred lytic reactivation is kept low so as to balance cellular materials both for cell survival and the extra work on viral proteins production. Chemically reactivating EBV lytic cycle by C7 or SAHA may disrupt the homeostasis and result in the imbalance of cellular molecules towards extensive EBV lytic protein expression, consequently reducing cellular materials for sustaining cell survival and eventually resulting in cell death. Given that these lytic inducers reactivate EBV lytic cycle via different mechanisms and imposed distinctive cellular effects, we hypothesized that combining inducers from each of the subclass would enhance the percentage of cells undergoing lytic cycle, thus augment the killing of EBV+ NPC cells. In order to test this hypothesis, we treated HA, C666-1 and NPC43 cells with C7 and SAHA simultaneously and measured their combinatorial effects via cell proliferation, degree of DNA fragmentation and the expression of Zta, cell death and apoptotic markers by flow cytometry.”
Page 15 line 437-445: “From the above section, despite the enhanced cell death found in cells treated with the combination of C7 and SAHA, unexpected reduction in Zta expression level was resulted. In addition, only 15% of cells on average expressed both Zta and cell death marker, indicating that the majority of cell death was due to cellular mechanism unrelated to EBV lytic reactivation. This resulted in non-specific killing of both EBV+ and EBV- NPC cells. Indeed, synergism in cell death was resulted in HONE-1 cells when treated with the combination of C7 and SAHA (Figure S7). In order to focus the killing specifically to EBV+ NPC cells, we aimed to enhance the proportion of cells undergoing EBV lytic reactivation by matching their kinetics for lytic reactivation at the lowest dosage and shortest available treatment duration of C7 and SAHA.”
Supplementary Figure 7.
Page 21 line 711: “Figure S7. Non-specific killing in both EBV+ and EBV- NPC cells. HONE-1 cells were treated with a gradient combination of C7 and romidepsin for 48 hrs. Synergisms of proliferation inhibition of the two drugs in these cells were analyzed by isobologram analysis.”
Page 15 line 466-468: “Moreover, cells expressing both Zta and cell death marker has increased to 23%, suggesting that with this strategy, focused EBV+ cell death could also be achieved.”
Page 21 line 695-702: “In conclusion, this study shows for the first time that autophagy initiation, in particular, the ATG5 protein is required for EBV lytic reactivation in NPC. C7/iron chelators and HDACi induce autophagy-dependent and independent mechanisms, respectively, to reactivate lytic cycle of EBV while imposing differential cellular effects. Lastly, combination of C7 and SAHA at their corresponding reactivation kinetics enhanced EBV lytic reactivation. This study could provide the field with new insights on EBV pathogenesis by understanding the mechanism regulating EBV latent-lytic switch and stimulate a rational design in drug combinations against EBV-associated cancers.”
Comment:
“How do they confirm abortive EBV lytic infection in various autophagy modulator treated experiments? The early and/or late EBV gene expression was not determined in the paper.”
Response:
Thank you for your question. We have shown previously in (https://doi.org/10.1371/journal.pone.0145994) and (doi: 10.3390/cancers10120505.) by western blot and immunofluorescent staining that late lytic proteins could not be found in C7 treated cells. Moreover, Raji cells could not be infected with supernatant collected from EBV+ HA cells induced by C7 in a transfection assay Therefore, we believed that C7 reactivates an abortive EBV lytic cycle.
In this manuscript, given with the results in Figure 2a, Zta expression has decreased to 0% when the cells were treated in the presence of 3-MA, without Zta, we therefore believed that there was no lytic reactivation. As in the case of CQ, we have unpublished data showing enhanced expression of Zta, Rta and EA-D in C7-treated cells while the expression the above IE and E lytic genes were similar in cells treated with the combination of C7 and CQ. For the late lytic genes gp350, in cells treated with C7 or with combined C7 and CQ, their expression was lower than that in the untreated cells; while for VCA-p18, the expression for all of the three conditions were the same, therefore we believed that the addition of CQ would not affect the original effect of C7 imposed on the cells. We therefore believed that C7 induced an abortive EBV lytic cycle regardless of the presence of autophagy modulators.
Comment:
“What the cell death mechanism that is not mediated via apoptosis? How the autophagy modulation the authors used cause these NPC cell killing.”
Response:
Thank you for your question, we believed that autophagic cell death was involved in addition to apoptosis. In cells treated with the combination of C7 and SAHA, the apoptotic marker cleaved caspase-3 did not increased with the enhanced cell death, we therefore conjectured that in addition to apoptosis, other cell death mechanism was involved and that could be autophagic cell death as higher autophagy expression could be observed in cells treated with the combined drugs. We believed that when the cell is highly stressed by the introduction of both of the lytic inducers, autophagy will switched from its protective mode to its killing mode, consequently leading to autophagic cell death. To aid delivery of this message, we have made amendments on page 13 line 395-401.
Amendments:
Page 13 Line 395-401: “This suggested in addition to apoptosis, another cell death mechanism is involved in cells treated with the combined drug at 24 and 48 hrs. Given that enhanced accumulation of LC3B puncta was observed in cells treated with the combination of C7 and SAHA (Figure S4), we conjectured that cells were severely damaged due to the combined treatment and have converted autophagy from a protective mechanism to a suicide state, resulting in autophagic cell death. This potentially explained the non-apoptotic cell death enhancement observed in cells treated with the combined drug.”
Comment:
“How do we apply the result in the understanding of NPC carcinogenesis or in the NPC treatment?”
Response:
Thank you for your question. We believed that the identification of the predominant role of ATG5 in EBV lytic reactivation could provide new insights to the field on EBV pathogenesis in understanding cellular factors that control EBV latent-lytic switch, allowing us to learn the basic biology of EBV in manipulating a well-conserved cellular mechanism.
The final combination with matching kinetics for lytic reactivation at the lowest dosage and shortest available treatment duration of C7 and SAHA has enhanced the proportion of cells undergoing lytic reactivation. Moreover, cells expressing both Zta and cell death marker has increased to 23%, suggesting that with this strategy, specific EBV+ cell death could be achieved. We believed that this could stimulate a rational design in drug combinations against EBV-associated cancers.

Reviewer 3 Report
This is a nice and sound piece of work; the experiments are well designed and presented. The manuscript reads well and despite the link between autophagy and EBV lytic reactivation was already shown in B-cells, no previous studies explored this issue in epithelial cells (NPC). Importantly this study tests in controlled in vitro models the effect of different compound with the potential to induce cell death of EBV+ NPC cells. I therefore support the publication of this manuscript in this Journal.
Minor comments:
1) Previous literature on the ATG5 role in EBV lytic activation should be more thoroughly cited (e.g. DOI: 10.1128/JVI.02033-14)
2) I will change the expression “particularly required” (at pag5 line 153 and page 6 line 194) as it is too vague. Please be more precise: exclusively? or plays an important role in? is necessary but not sufficient?
3) Page 5 line 163 please replace “the about result” by ‘the above result”
4) In the discussion section the authors claim that ATG5 is down-regulated in NPC (page 16 line 451), and further down-regulated upon siRNA. This assumption is based on the evidence that ATG5 is down-regulated in some cancers. Have the authors got other evidence to support this statement? Actually when looking at some of the western blots images, ATG5 basal levels appear to be quite high in EBV+ NPC cell lines.
5) Is it known whether expression of BZLF1 alters autophagy pathway associated gene expression? Could this information be recovered by in silico analysis of published data. Could the authors comment on this possibility?
Author Response
Response to reviewer 3’s comments:
Comment:
“Previous literature on the ATG5 role in EBV lytic activation should be more thoroughly cited (e.g. DOI: 10.1128/JVI.02033-14)”
Response:
Thank you very much for your suggestion, amendments have been made on page 2, line 83-87
Amendments:
Page 2 Line 83-87: “In EBV lytic cycle induction, a study showed that the immediate early lytic protein Rta but not Zta, could initiate autophagy via the ERK1/2 signaling pathway. By inhibiting autophagy by 3-MA and atg5 knockdown abrogated the expression of EBV lytic proteins and the production of viral particles in B cells”
Comment:
“I will change the expression “particularly required” (at pag5 line 153 and page 6 line 194) as it is too vague. Please be more precise: exclusively? or plays an important role in? is necessary but not sufficient?”
Response:
Thank you for your comment, amendments have been made” on page 5 line 160-161 and page 6 line 203-204.
Amendments:
Page 5 line 160-161 “Autophagic protein ATG5 is predominantly required for EBV lytic reactivation upon C7 treatment”
Page 6 line 203-204 “Autophagic protein ATG5 is predominantly required for EBV lytic reactivation upon C7 treatment”
Comment:
“Page 5 line 163 please replace “the about result” by ‘the above result”
Response:
Thank you for your reminder, it has been amended, please see page 5, line 171.
Amendments:
Page 5 line 171: ‘The above result prompted us to hypothesize that…”
Comment:
“In the discussion section the authors claim that ATG5 is down-regulated in NPC (page 16 line 451), and further down-regulated upon siRNA. This assumption is based on the evidence that ATG5 is down-regulated in some cancers. Have the authors got other evidence to support this statement? Actually when looking at some of the western blots images, ATG5 basal levels appear to be quite high in EBV+ NPC cell lines.”
Response:
Thank you for pointing it out, we do not have other evidence to support that ATG5 is downregulated in EBV+ NPC cells. The reason for this part is because autophagy is a complicated machinery, each of the autophagic proteins are interrelated to one another. It is true that in our case, the basal level of ATG5 is high and we agree that it seems not being down-regulated. However, we would like to be very cautious for not over-claiming our results and remind readers that despite ATG5 has a predominantly role in EBV lytic reactivation, we should not neglect the possible role of other autophagic proteins because in this manuscript, not every single autophagic protein is being examined. We have made amendments regarding this point, please see page 17 line 518-523.
Amendments:
Page 17 line 518-523: “Since autophagy is a complicated machinery and each of the autophagic proteins are interrelated to one another, and in this study, not every single autophagic protein is being examined, therefore, despite ATG5 is identified as a predominantly factor in EBV lytic reactivation in this context, we have to be cautious that the involvement of other autophagic proteins cannot be completely excluded.”
Comment:
“Is it known whether expression of BZLF1 alters autophagy pathway associated gene expression? Could this information be recovered by in silico analysis of published data. Could the authors comment on this possibility?”
Response:
It is not known in epithelial cells, but in the context of B cells, Rta has been shown to initiate autophagy. It would certainly be interesting to analyse the data in silico yet we would be skeptical because autophagy is very sensitive to the changes in the environment, which consequently affects the expression of autophagic genes. Published data derived from different studies may vary in their culturing system and the method for EBV lytic induction, which may confound the expression of autophagic genes.

Round 2
Reviewer 2 Report
The authors has revised the manuscript thoroughly. I think it is worth accepting.
Reviewer 3 Report
The majority of the comments have been addressed by the authors, so the manuscript is now acceptable for publication.